**Short summary.** 250 m estimates of snow water equivalent in the Western US and Canada are improved by assimilating observations representative of a snow-focused satellite mission with a land surface model. Here, by including a gap-filling strategy, snow estimates could be improved in forested regions where remote sensing is challenging. This approach improved estimates of winter maximum snow water volume to within 4%, on average, with persistent improvements to both spring snow and runoff in many regions.

# Extending the utility of space-borne snow water equivalent observations over vegetated areas with data assimilation

Justin M. Pflug[1,2], Melissa L. Wrzesien[1,2], Sujay V. Kumar[2], Eunsang Cho[1,2,a], Kristi R. Arsenault[3,2], Paul R. Houser[4], Carrie M. Vuyovich[2]

[1]Earth System Science Interdisciplinary Center, University of Maryland, College Park, MD, USA
[2]Hydrological Sciences Laboratory, NASA Goddard Space Flight Center, Greenbelt, MD, USA
[3]Science Applications International Corporation, McLean, VA, USA
[4]Geography and Geoinformation Science Department, George Mason University, Fairfax,VA, USA
[a]current address: Ingram School of Engineering, Texas State University, San Marcos, TX, USA

*Correspondence to*: Justin Pflug (Justin.Pflug@nasa.gov)

**Abstract.** Snow is a vital component of the Earth system. Yet, no snow-focused satellite remote sensing platform currently exists. In this study, we investigate how synthetic observations of snow water equivalent (SWE) representative of a synthetic aperture radar remote sensing platform could improve spatiotemporal estimates of snowpack. We use a fraternal twin Observing System Simulation Experiment, specifically investigating how much snow simulated using widely used models and forcing data could be improved by assimilating synthetic observations of SWE. We focus this study across a 24°-by-37° domain in the Western United States (US) and Canada, simulating snow at 250 m resolution and hourly timesteps in water-year 2019. We perform two data assimilation experiments, including: 1) a simulation excluding synthetic observations in forests where canopies obstruct remote sensing retrievals, and 2) a simulation inferring snow distribution in forested grid cells using synthetic observations from nearby canopy-free grid cells. Results found that, relative to a nature run, or assumed true simulation of snow evolution, assimilating synthetic SWE observations improved average SWE biases at maximum snowpack timing in shrub, grass, crop, bare-ground, and wetland land cover types from 14%, to within 1%. However, forested grid cells contained a disproportionate amount of SWE volume. In forests, SWE mean absolute errors at the time of maximum snow volume were 111 mm, and average SWE biases were on the order of 150%. Here, the data assimilation approach that estimated forest SWE using observations from the nearest canopy-free grid cells substantially improved these SWE biases (18%) and the SWE mean absolute error (27 mm). Simulations employing data assimilation also improved estimates of the temporal evolution of both SWE and runoff, even in spring snowmelt periods when melting snow and high snow liquid water content prevented synthetic SWE retrievals. In fact, in the Upper Colorado River region, melt-season SWE biases were improved from 63% to within 1%, and the Nash Sutcliffe Efficiency of runoff improved from –2.59 to 0.22. These results demonstrate the

value of data assimilation and a snow-focused globally relevant remote sensing platform for improving the characterization of
SWE and associated water availability.
**1 Introduction**
Snow plays important roles in the Earth system by regulating global temperatures and cooling the land surface because of
its reflective properties (Barry, 2002). Snow is also a major source of water storage for many regions, especially in areas that
rely on snowpack to sustain water resources during the dry season. In fact, it has been estimated that more than 2 billion people
around the world are reliant on seasonal snow melt for their water supply (Barnett et al., 2005). Snowpack is the natural
'integrator' of climatic conditions and offers more predictability of water availability than variables with shorter memory, such
as precipitation and streamflow (Terzago et al., 2023). Accurate wintertime estimates of snowpack are therefore critical for
water management and agricultural planning (Koster et al., 2010). For example, in the Western US, where a vast majority of
streamflow originates from snow (Li et al., 2017), it is common practice to use the April 1 snowpack, the historic date of
maximum snowpack in that region, for developing water supply estimates for later in the season. However, climate change
impacts have led to increased variability in the snow seasonality (Livneh and Badger, 2020), with warmer temperatures
reducing the amount of snow accumulation and seasonal snow storage, and advancing the timing of the spring melt. Therefore,
accurate characterization of winter snowpack and its variability is critically important for making informed water supply
quantifications.
In recognition of the critical need to have spatially distributed measurements of snow mass, there have been several efforts
to measure and estimate SWEfrom many different remote sensing platforms in the past several decades. Airborne lidar systems
have been able to provide high resolution, accurate measurements of snow mass (Painter et al., 2016), but this approach has
significant logistical barriers for global and frequent snow measurements, and the hydrological utility of a practical spaceborne
lidar platform is limited (Kwon et al., 2021). In the past three decades, snow depth and SWE estimates have been derived from
passive microwave remote sensing measurements, but these measurements are at coarse spatial resolutions, and have limited
accuracies over deep and wet snow, complex terrain, and dense vegetation (Derksen et al., 2014; Foster et al., 2005). Active
microwave remote sensing instruments such as Synthetic Aperture Radars (SARs) can provide finer spatial resolution
measurements to help resolve some of these issues. For example, C-band SAR observations from the Sentinel-1 constellation
have shown promise in obtaining high quality, moderate resolution (1km) observations in deep snow environments (Lievens
et al., 2019). A volume scattering radar approach, using X- and Ku-band SAR, has also been demonstrated in several airborne
campaigns and proposed for multiple snow mission concepts (Yueh et al 2009, Rott et al 2010) because of its potential to
achieve high resolution and global coverage over a range of snow depths. While these microwave instruments can observe in
night-time and cloudy conditions, they are still limited over areas with dense vegetation (Tsang et al., 2022). Further, all
spaceborne instruments have inherent coverage gaps due to their orbital and revisit configurations.

To overcome these limitations, modeling and data assimilation systems are needed that can extend the coverage and utility of available measurements to areas, times, and variables that are not directly observed. In this article, we present a novel approach through data assimilation, designed specifically to improve the usefulness of spaceborne SWE retrievals over forested areas. The approach is demonstrated using an observing system simulation experiment (OSSE; e.g., Cho et al., 2023; Errico et al., 2007) which is an approach used to formally assess the impact of the data to be collected from an anticipated mission. Several prior studies have examined the use of OSSEs for snow mission studies (Garnaud et al., 2019; Kwon et al., 2021; Wrzesien et al., 2022). Among them, SAR-focused OSSEs have been conducted by Garnaud et al. (2019) and Cho et al. (2023) to assess the utility of hypothetical snow observations. Garnaud et al. (2019) focused on a Ku-band SAR to quantify trade-offs between sensor configurations (e.g., various spatial resolutions and revisit frequencies) with retrieval algorithm accuracy and SWE performance in southern Quebec, Canada, where temperate forests are dominant with shallow and moderate snowpack conditions. Cho et al. (2023) conducted a X-/Ku-band SAR OSSE with an achievable sensor configuration (1 km spatial resolution, 7-day revisit frequency, and orbital configurations) focusing on mountainous environments in a western Colorado and testing the degree to which various SAR retrieval capabilities in different forest densities and snow volumes could improve observationally-based SWE estimates. Here, we build on this prior research by developing an OSSE covering the entire western US and portions of Canada. We simulate finer-scale (250 m) synthetic SWE observations that could be provided from a future X-/Ku-band SAR mission, which are then incorporated within a land surface model through data assimilation to assess their capability to improve snow state estimates, and the integrated impact on hydrologic states in space and time. The assimilation experiments here are conducted with and without a novel strategy to extend SAR-based SWE estimates from unforested regions into forested landscapes where SAR retrievals of the snowpack may be obscured by the forest canopy.

The primary contribution of this paper is the development of a viable strategy for extending hypothetical remotely sensed SWE retrievals from a volume-scattering X-/Ku-band SAR satellite mission into difficult-to-observe forest landscapes. We specifically focus on addressing the following research questions: 1) what is the added utility of spaceborne active remote sensing SWE retrievals (assuming retrievals meet currently defined mission requirements) across the Western US and Canada? 2) how much can spatiotemporal representations of SWE be improved by focusing on developing observationally based snow estimates over areas with dense vegetation, where SAR sensors may be limited? 3) How much added hydrological utility can be obtained through spaceborne active remote sensing measurements and data assimilation approaches, particularly when coverage over forested areas is improved?

Section 2 describes the study domain and OSSE modeling setup. This is followed by the description of the results (Sect. 3), a discussion of the findings (Sect. 4), and the study's conclusions (Sect. 5).

**2.1 Study domain and OSSE setup**

An OSSE is used to assess the value of the data to be collected from an anticipated mission. OSSEs often consist of the following steps: 1) Developing a "nature run" that uses a state-of-the-art model employed with the best available boundary conditions (Sect. 2.1); 2) using the nature run to generate simulated remote sensing observations, accounting for sources of sensing limitations, sensing uncertainties, and orbital configurations (Sect. 2.2); 3) incorporating the simulated observations (often through data assimilation, Sect. 2.3) in a separate, "open loop" model configuration with accuracies representative of common modeling biases and uncertainties; and 4) evaluating how much the simulated remote sensing data improve the open loop model performance relative to the nature run. In addition to this OSSE approach, this study goes further by 1) testing the degree of improvement to both the remotely-sensed variable (i.e., SWE) and the resulting changes to land surface runoff in snow covered regions, and 2) developing two separate data assimilation experiments, one which masks simulated observations in forested pixels where SAR retrievals may be most challenging, and the other including a novel approach for inferring SWE in forested pixels using simulated observations from nearby, unforested pixels (Sect. 2.4). The details of the OSSE setup used in this study are described in more depth throughout this section.

We employ the NASA Land Information System (LIS; Kumar et al., 2006), an infrastructure for high performance, ensemble-based land surface modelling and data assimilation to enable this OSSE. LIS encompasses several advanced land surface models that can simulate terrestrial water, energy, and carbon balances and related states such as soil moisture, land surface temperature, and SWE, among others. These include different versions of community models such as Noah (Ek et al., 2003), Variable Infiltration Capacity (VIC; Liang et al., 1994), Catchment (Koster et al., 2000), Joint UK Land Environment Simulator (JULES; Best et al., 2011), and Noah-MP (Niu et al., 2011). The LIS framework also includes support for specialized models that are designed to provide more detailed representations of certain land surface processes (e.g. snow), while enabling interaction with LSMs that solve for water, energy, and carbon balances at a macroscale. For example, the advanced snow physics model called SnowModel (Liston and Elder, 2006) has been incorporated within LIS in a manner that allows coupling to existing LSMs. This structure allows the use of the advanced snow physics from SnowModel while leveraging the existing process schemes (e.g., sub-surface, groundwater, canopy) within the LSMs. Here we utilize these unique capabilities for enabling the OSSE integrations. . The study is conducted over a large domain (Fig. 1), covering the Western US and southern Canada from 31-55N and 93-130W at a 250 m spatial resolution. As shown in Fig. 1, this modeling domain encompasses a broad range of vegetation types, topographical regimes and water resources regions of the Pacific Northwest, California, Great Basin, and Upper Colorado. 22% of the domain is covered by forests, with grasslands, croplands, and shrublands accounting for 20%, 23%, and 26% of the domain, respectively. Forests dominate the coverage of areas with significant snowpack, occupying 58% of regions that are in the mid-elevation range of 2500-3500m, and 15% of the areas with elevations over 3500m. From a modeling perspective, the domain extent of Fig. 1 (~83 million land grid points) is computationally challenging. The scalable high performance computational and parallel inputting and outputting capabilities of NASA LIS were leveraged to enable these simulations. A multiprocessor configuration involving approximately 1000 processors was employed to facilitate large model simulations for the nature run, open loop simulation, and two simulations with data assimilation.

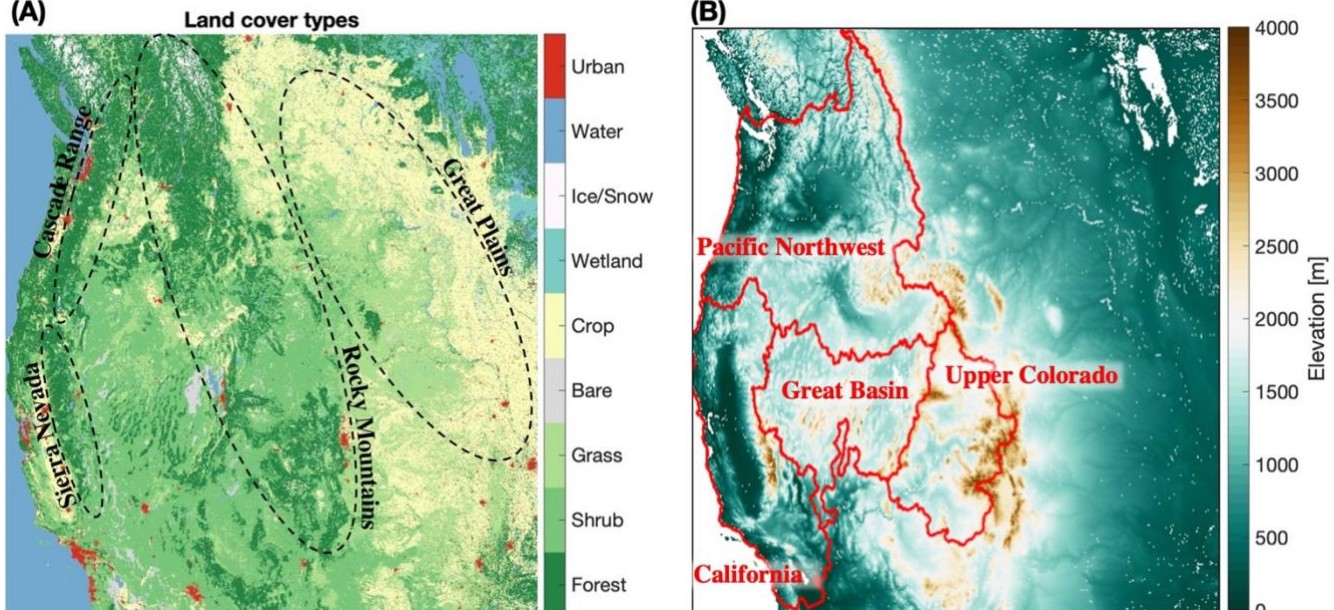

**Figure 1:** Maps of the land and vegetation classes (A; left panel) and elevation (in meters) (B; right panel) used in the
simulations. Outlines and labels in the left panel indicate regions discussed in the Results (Sect. 4). Red contours in the right
panel indicate hydrologic regions used in the analysis.
Simulations in this study are conducted by forcing LIS LSMs with the surface meteorology from NASA's Modern Era
Retrospective Reanalysis, version 2 (MERRA-2; Gelaro et al., 2017) and ECMWF Reanalysis, version 5 (ERA5; Hersbach et
al., 2020) products. The model integrations were conducted for the water year 2019 (September 2018 – September 2019),
which was a wetter than normal year based on the long-term average meteorological conditions over this domain.
The open loop and data assimilation integrations performed in this study are conducted using the Noah land surface model
with multi-parameterization (Noah-MP) version 4.0.1 (Niu et al., 2011) and forcing from ERA5.The Noah-MP model evolved
from the Noah LSM, with multiple options for various land surface processes. It represents energy, water, and carbon balances
at the land surface by accounting for processes related to infiltration, evaporation, transpiration, runoff generation and
groundwater recharge. A TOPMODEL-based runoff model (Beven et al., 2021) is used to calculate surface runoff and
groundwater discharge. Options for prognostic vegetation dynamics models that represent the growth and senescence of
vegetation are also available within Noah-MP. A two-stream radiative transfer approach is employed to calculate surface
energy processes. Finally, a multilayer snowpack model (with up to three layers) is used to account for snow melt
metamorphisms, compaction by overlying snow, sublimation of canopy intercepted snow, and snowmelt-refreeze cycles within
Noah-MP (Niu and Yang, 2004).
Snow states like snow depth and SWE were also modeled across the Western US (domain highlighted in Fig. 1) at 250 m
resolution and hourly time steps using a state-of-the-art and physically based single-layer snow model(named SnowModel;
Liston and Elder, 2006), provided forcing from MERRA-2 with LIS-provided lapse rates and topography-based meteorological
downscaling approaches, like incoming shortwave corrections based on topographical shading (Cosgrove et al., 2003; Kumar
et al., 2013). SnowModel has seen widespread use in the snow community, demonstrating the capability to resolve snow
evolution in a variety of landscapes and complex snow processes like the redistribution of snow via wind, and the resulting
impact on snow distribution, melt season snow duration, glacier mass balance, and snow habitat for species like polar bears
and Dall sheep (Hiemstra et al., 2002; Liston et al., 2016; Mahoney et al., 2018; Mernild et al., 2017; Sturm and Wagner,
2010). In addition to wind redistribution, snow evolution within SnowModel accounts for a wide set of snow processes,
including snow sublimation, snow grain size evolution, solar topographical shading, canopy shading, and canopy snow
interception. Through the coupling within LIS, Noah-MP snow states and the resulting snow-driven runoff were updated using
the SnowModel outputs at hourly timesteps for each grid cell.
Preliminary research has shown that relative to Noah-MP, LIS simulations coupling Noah-MP with SnowModel have
improved the volume and spatial distribution of simulated snow depth and SWE (Arsenault et al., 2021; Wrzesien et al., 2022).
Therefore, the coupled SnowModel and Noah-MP model was a prime candidate for the "nature run" in this study, or the
simulation most representative of the true underlying spatiotemporal snow states from which simulated observations were
derived (Sect. 2.2), and the assimilated model was compared against. Here, the nature run and open loop simulations detailed
above were compared to a widely-used Western US snow reanalysis product (Fang et al., 2022) to ensure that 1) the nature
run exhibited reasonable model accuracy, and 2) the departure between the open loop simulation and nature run are
representative of common regional, continental, and global modeling efforts (Figure S1 and S2). The OSSE developed for this
study is a "fraternal twin" OSSE, wherein two different models are used to simulate snow in the open loop (Noah-MP) and
nature run (SnowModel) simulations. This approach is selected since "identical twin" OSSEs, which use the same model, can
result in less divergence in model states and information content, biasing the degree of model improvement that could come
from assimilating an observation (e.g., Yu et al., 2019). More information on the difference between the open loop and nature
run models can be found in Table S1.
**2.2 Observation simulator**
. Synthetic SWE retrievals at 250 m spatial resolution, representative of a hypothetical X- and Ku-band SAR mission, are
simulated from the nature run. To do this, the orbital swaths are simulated using TAT-C (Le Moigne et al., 2017). TAT-C is a
NASA software system designed for future Distribution Spacecraft Missions (DSM), which enables us to explore a range of
feasible design options (e.g., constellation vs. single, geostationary vs. polar-orbiting, low vs. high temporal frequencies) to
estimate optimal gains for the given mission configuration. Previous OSSEs have been conducted to test the impact from
different snow mission configurations (e.g. Garnaud et al 2019). Here we instead focus on demonstrating the value of a gap-
filling approach (Sect. 2.4) for estimating snow in forested landscapes where SAR retrievals may be most challenging.
Therefore, we used TAT-C to design a conservative mission configuration consisting of a small constellation of X- and Ku-

band SAR satellites. Using a 10-14 day revisit frequency, depending on latitude, TAT-C orbital swaths were applied to the nature run outputs to simulate the satellite viewing area. The remote sensing spatial coverage is simulated by extending the ground track to a swath width. The daily viewing extents are then simulated as a daily binary map (so-called "cookie cutter") masking the surface as viewed or not at a 250 m spatial resolution.

Based on an error level of 20%, spatially and temporally uncorrelated random errors drawn from a Gaussian distribution are added to the synthetic SWE retrievals. This 20% error level is selected using a conservative estimate of SWE measurement uncertainty for a volume-scattering X-/Ku -band SAR mission based on developed mission design concepts and ground validation. For example, the ESA Cold Regions Hydrology High-Resolution Observatory (CoREH2O) mission expected to meet instrument and retrieval requirements of $\pm 30$ mm accuracies for SWE of 300 mm, $\pm 10\%$ for SWE greater than 300 mm (Rott et al 2010, 2012). Similarly, the Canadian Terrestrial Snow Mass Mission (TSMM) concept that is currently under development aims to achieve better than 20% measurement uncertainty for SWE greater than 50 mm, though it is expected to have higher uncertainties in deep snow conditions (e.g., $\geq 200$ mm SWE)(Garnaud et al. 2019). Airborne and tower-based field data have demonstrated that a combination X- and Ku-band system can provide SWE retrievals over a range of snow conditions at accuracies better than 20% (Zhu et al. 2018, 2021, Tsang et al 2022, Durand et al. 2023, Singh et al. 2023). However, we use an assumption of uniform error levels throughout the domain, whereas in reality, the errors are likely to be dependent on other factors, including the terrain characteristics, snow characteristics, and vegetation. This is discussed more in Section 4.

**2.3 Data assimilation setup**

A one-dimensional ensemble Kalman Filter (EnKF; Reichle et al., 2002) is used to assimilate the synthetic observations within the open loop configuration of the model. EnKF is widely used for land data assimilation studies (Kumar et al., 2022), as it provides a flexible approach for the treatment of model and observation errors and non-linear models. An ensemble of model realizations is used by EnKF to assess and propagate model errors. In this instance, the ensemble requirement further adds to the significant computational requirements of the large model domain (Fig. 1) and fine spatial resolution of the simulations (250 m). Therefore, a 5-member ensemble with perturbations applied to the meteorological variables and model prognostic fields are used for simulating uncertainty in the modeled estimates. Table 1 details the parameters for meteorological and model state perturbations, which are based on recent snow data assimilation studies (Lahmers et al., 2022; Kwon et al., 2021). Though a larger ensemble size is better for ensuring sufficient sampling density, our choice of five ensembles is reasonable given that the model state vector used in the assimilation only consists of two variables; the total SWE and snow depth. The assimilation setup employs a sequential update strategy, where at each time step an ensemble of model forecasts is propagated forward in time, followed by an update based on observational inputs. The model states are updated toward the observations based on the relative uncertainties in the model and observations using the following formulation, at a certain time $k$.

$$x_k^{i+} = x_k^{i-} + K_k\left[y_k^i - H_k x_k^{i-}\right] \qquad\qquad \text{Eq. (1)}$$


Where $x_k$ and $y_k$ are the model and observation state vectors, respectively. The term $H_k$ represents the observation operator
that maps the model states to the observed variables. The superscripts $i-$ and $i+$ represent the $i$th ensemble member before
and after the update, respectively. $K_k$ is the "Kalman gain" term, that allows the weighting of the observations and model
forecasts is a function of the model and observation error covariances.
**Table 1.** Model forcing and state-variable perturbations used by the 5-member ensemble of LIS simulations

| Variable | Perturbation Type | Std. Dev. | Cross Correlation across variables | | | |
|---|---|---|---|---|---|---|
| Meteorological Forcing | | | SW corr | LW corr | PCP corr | T corr |
| Downward Shortwave (SW) | Multiplicative | 0.2 | 1 | −0.3 | −0.5 | 0.3 |
| Downward Longwave (LW) | Additive | 30 | −0.3 | 1 | 0.5 | 0.6 |
| Precipitation (PCP) | Multiplicative | 0.5 | −0.5 | 0.5 | 1 | −0.1 |
| Near surface Air Temperature (T) | Additive | 0.5 | 0.3 | 0.6 | −0.1 | 1 |
| Noah-MP LSM snow states | | | SWE | Snow depth | | |
| SWE | Multiplicative | 0.01 | 1 | 0.9 | | |
| Snow depth | Multiplicative | 0.01 | 0.9 | 1 | | |

The data assimilation procedure detailed here assimilated the synthetic SWE retrievals (Sect. 2.2) with the open loop
simulation. The degree to which the simulation with data assimilation approached SWE simulated by the nature run is intended
to represent the extent to which a SAR remote sensing platform with the SWE retrieval characteristics from Sect. 2.2 could be
combined with a land surface model to provide near real-time estimates of SWE at 250 m resolution. However, the SAR
observations synthesized in this study have known issues with observing snow with high liquid water contents and dense forest
cover. Therefore, synthetic observations at each timestep were masked at grid cells where the most-dominant landcover type
from the North American Land Change Monitoring System (NALCMS; Latifovic et al., 2017) was forested, including
deciduous, evergreen, and mixed forest cover (Fig. 1). Synthetic observations were also masked at grid cells where and when

snow was experiencing melt, identified by the presence of liquid water in the snowpack from the nature run. Although limited in area, grid cells with "ice" landcover (Fig. 1) were also excluded. In this study, this simulation which used assimilation only in unforested, non-melting, and ice-free grid cells is termed "Data Assimilation, without the forest strategy" (DA). In Sect. 2.4 below, we present a novel approach used to infer SWE in grid cells with forests using the nearest canopy-free synthetic observations.

**2.4 Extending observations over forests**

The 1-d EnKF approach employed here updated each model grid SWE from the open loop simulation based on the observations available at that grid point. Though studies have employed 3-d EnKF approaches to spatially propagate observational information to neighbouring grid cells (De Lannoy et al., 2012), here we relied on 1-d updates due to several factors. First, a 2-d update requires the knowledge of spatial error correlations and their variability, which is challenging to specify (Ying, 2020). Most prior studies using such schemes employ uniform specifications and are limited to small domains. Second, a 2-d update increases the size of the state vector and consequently requires the use of a larger ensemble. This, combined with the added computational expense of a 2-d analysis significantly increases the computational cost. Therefore, we employed an alternate approach that is computationally more efficient while allowing the extension of observations to nearby areas.

Assuming that the SWE retrievals from the hypothetical SAR instrument are limited over areas where the dominant vegetation type are forests (Fig. 1a), we employ a novel approach to extend the observations obtained in non-forested areas (Fig 2). For every forested location, valid retrievals over nearby non-forested locations within a radius of influence of 750 m are identified. An observation at the forested pixel is then estimated by scaling the model SWE by the ratio of the average observed SWE to modeled SWE over the 'clearing' areas (Fig. 2). This scaled observation is then used for assimilation over the forested pixel. Here we implicitly use the spatial correlations inherent in the model between forested and clearing areas to extend observational coverage over the clearing to forested locations. This simulation is termed "Data Assimilation, with the forest strategy" (symbolized by DA+F in Section 3). To evaluate the accuracy and added value of this scaling approach, we compare SWE and runoff from the nature run simulation, versus simulations with data assimilation both 1) employing the forest scaling strategy discussed here, and 2) masking synthetic observations in forested grid cells (Sect. 2.3).

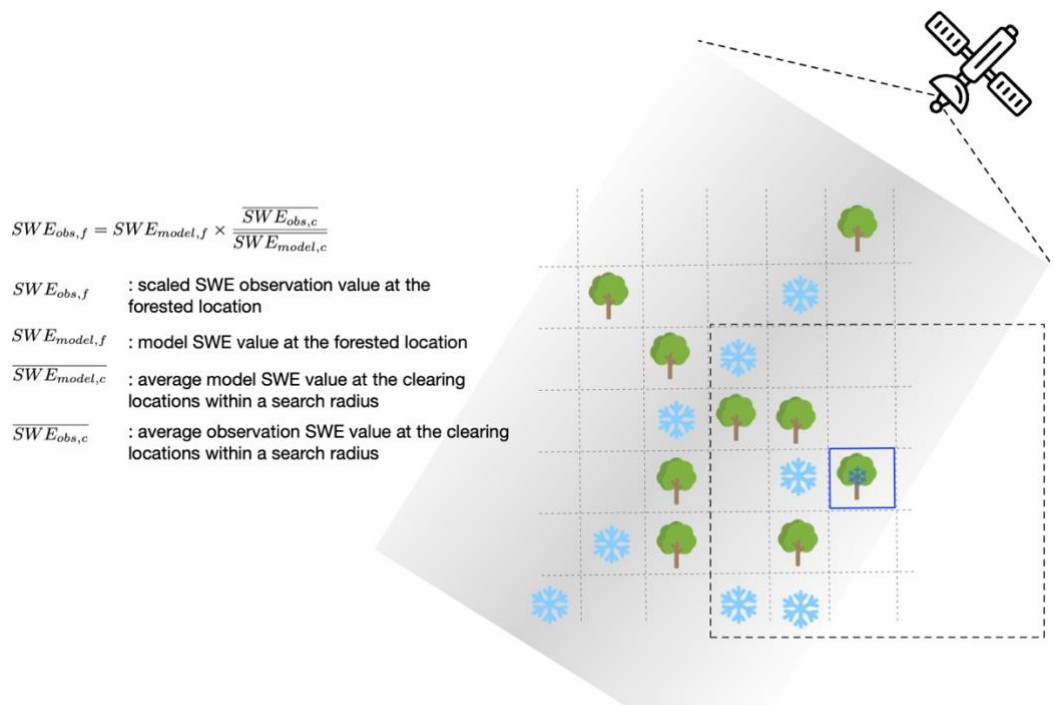

$$SWE_{obs,f} = SWE_{model,f} \times \frac{\overline{SWE_{obs,c}}}{\overline{SWE_{model,c}}}$$

$SWE_{obs,f}$ : scaled SWE observation value at the forested location

$SWE_{model,f}$ : model SWE value at the forested location

$\overline{SWE_{model,c}}$ : average model SWE value at the clearing locations within a search radius

$\overline{SWE_{obs,c}}$ : average observation SWE value at the clearing locations within a search radius

253

**Figure 2.** Conceptual depiction and equations demonstrating the forest strategy used here, which estimates a SWE observation at a given grid cell (outlined box in blue color) based on the modeled SWE ($SWE_{model,f}$) and the ratio between the average synthetic SWE observations ($\overline{SWE_{obs,c}}$) and average modeled SWE ($\overline{SWE_{model,c}}$) from grid cells within a 750 m radius (dashed box). The light gray shading represents the satellite swath, the tree icons indicate forested locations, and the snowflake icons represent grid cells with valid SWE retrievals at non-forested locations. The grid cell from this example is near the satellite swath edge, so observations are unavailable in the nearby regions South and East of this pixel.

## 3 Results

In this section, we compute the difference between the open loop simulation, nature run, and the two open loop simulations with data assimilation, one masking synthetic observations over regions with forests, and time periods with melting snow, and ice, and the other applying the same data assimilation but extending snow estimates in forested regions using the strategy from Sect. 2.4 and Fig. 2. The differences between these simulations are detailed in Section 2 and Table S1. We focus on the differences between these four simulations using: 1) average SWE from the winter snow accumulation season (December, January, and February; DJF), when snowmelt is minimized and synthetic observations are masked by grid cells with liquid water content to the smallest degree, 2) spatially distributed SWE on 13 March, the date corresponding to the timing of maximum SWE volume in water-year 2019, and 3) daily average SWE and total runoff for each day in water-year 2019 over a number of selected hydrologic regions including the Pacific Northwest, California, Great Basin, and Upper Colorado (Fig. 1b).

The open loop and nature run simulations exhibited differences in both the volume and spatial distribution of average winter (December, January, and February; DJF) SWE (Fig. 3a and 3b). Relative to the nature run, the open loop simulation tended to simulate lower elevation winter SWE that was both larger in magnitude and persisted for longer before melting. In the Pacific Northwest domain (Fig. 4), DJF average snow cover (defined as grid cells with mean DJF SWE exceeding 5 mm), was approximately 12% larger for the open loop simulation than the nature run (Table 2). These snow extent biases were also apparent in the other hydrologic regions (Figs. S3 – S5), where open loop snow extents exceeded snow extents from the nature run by 26% in the Upper Colorado, 45% in the Great Basin, and 6% in California. Visually, the nature run had significant increases in the spatial variability of winter SWE, better representing the differences in SWE between mountain peaks and valleys, and the patchiness of snow cover in regions with winter snowmelt and ephemeral snow cover (e.g., Fig. 4, Fig. S1). Relative to the nature run, DJF SWE from the open loop simulation was biased high across the full modeling region (Fig. 3) by approximately 26%, on average, with a mean absolute error of 41 mm and spatial coefficient of correlation of approximately 0.74. Across the Pacific Northwest (Fig. 4), DJF mean SWE biases were approximately 37%, with a mean absolute error of 55 mm. Open loop model performance for the other hydrologic regions can be found in Table 2.

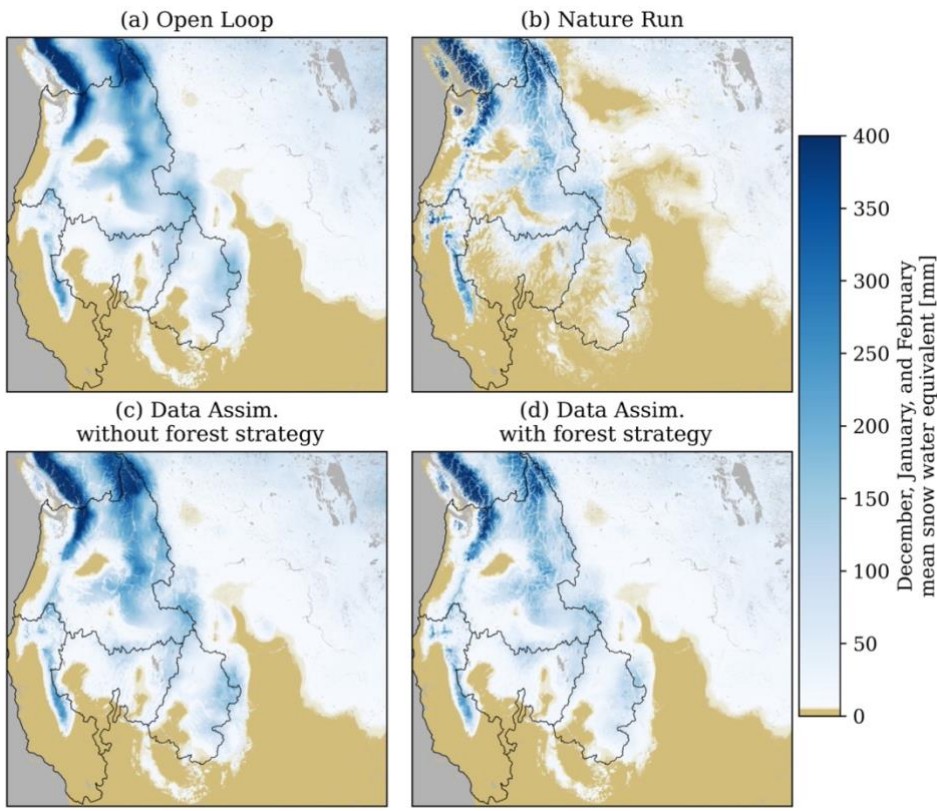

**Figure 3.** Winter (December, January, and February) mean SWE simulated at 250 m resolution from the open loop (a), nature run (b), and data assimilation simulations, both with (d) and without (c) the forest strategy presented in Sect. 2.4.

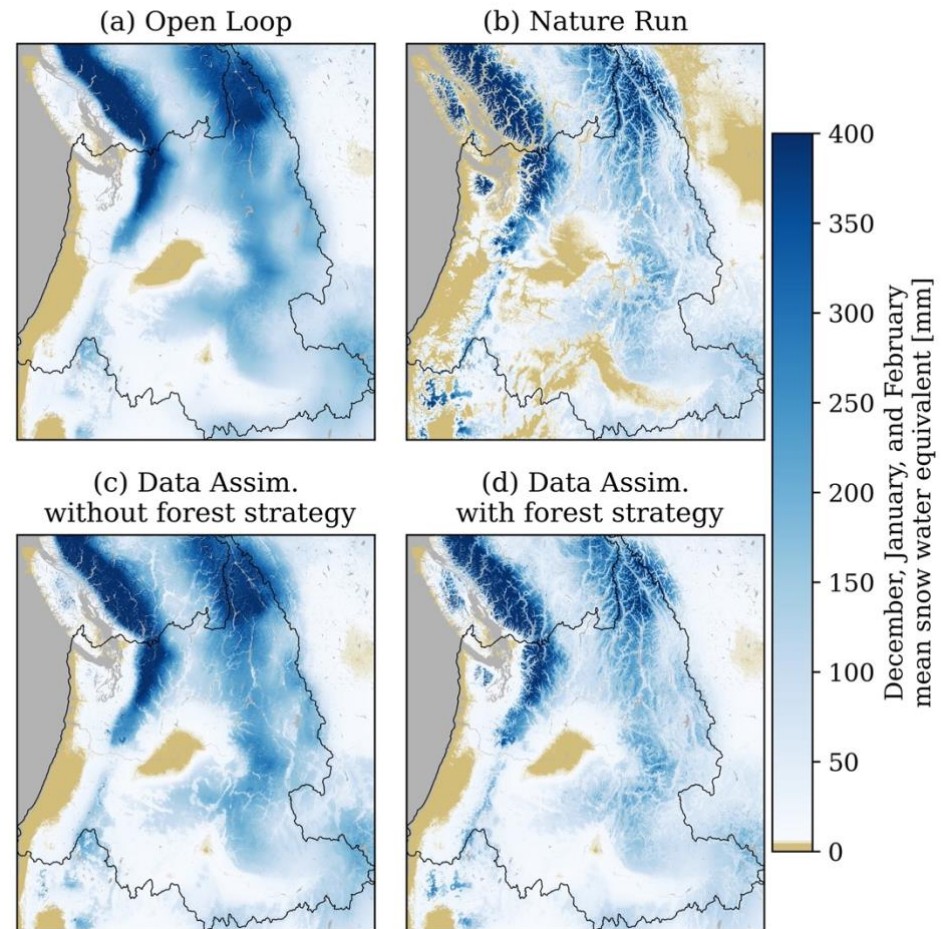

**Figure 4.** Winter (December, January, and February) mean SWE in the Pacific Northwest region simulated at 250 m resolution from the open loop (a), nature run (b), and data assimilation simulations, both with (d) and without (c) the forest strategy.

**Table 2.** Simulation performance, relative to the nature run simulation, for the open loop simulation (OL) and the simulations with data assimilation, both with (DA+F) and without (DA) the forest strategy. Statistics are presented for the full domain, the four hydrologic regions, and all forested and unforested grid cells.

| | | DJF* snow-extent biases | 13 March 2019 SWE | | | Seasonal SWE and runoff | |
| --- | --- | --- | --- | --- | --- | --- | --- |
| | | | Mean bias | SWE abs. error [mm] | Coeff. of corr. | MAM* mean SWE bias | Nash-Sutcliffe Efficiency |
| **Full study domain** | OL^ | +22% | +26% | 41 | 0.74 | - | - |
| | DA^ | +23% | +9% | 36 | 0.79 | - | - |
| | DA+F^ | +22% | +4% | 17 | 0.91 | - | - |
| **Upper Colorado** | OL | +26% | +37% | 55 | 0.74 | +63% | -2.59 |
| | DA | +28% | +27% | 50 | 0.74 | +86% | -3.71 |
| | DA+F | +28% | +8% | 23 | 0.90 | < 1% | 0.22 |
| **Pacific Northwest** | OL | +12% | +42% | 89 | 0.69 | +44% | -0.17 |
| | DA | +13% | +32% | 80 | 0.74 | +80% | -0.34 |
| | DA+F | +13% | +6% | 35 | 0.89 | +15% | 0.39 |
| **Great Basin** | OL | +45% | +35% | 38 | 0.62 | -29% | 0.58 |
| | DA | +46% | +46% | 32 | 0.75 | +10% | 0.58 |
| | DA+F | +46% | +28% | 23 | 0.83 | -38% | 0.53 |
| **California** | OL | +6% | -34% | 50 | 0.64 | -50% | 0.92 |
| | DA | +8% | -6% | 40 | 0.79 | -15% | 0.88 |
| | DA+F | +8% | -6% | 28 | 0.88 | -26% | 0.89 |
| **Unforested** | OL | +19% | +14% | 22 | 0.83 | - | - |
| | DA | +20% | < 1% | 14 | 0.91 | - | - |
| | DA+F | +20% | < 1% | 14 | 0.91 | - | - |
| **Forested** | OL | +29% | +150% | 111 | 0.67 | - | - |
| | DA | +30% | +150% | 111 | 0.67 | - | - |
| | DA+F | +30% | +18% | 27 | 0.93 | - | - |

*DJF = December, January, and February; MAM = March, April, and May (averages)
^OL = open loop simulation; DA = data assimilation without the forest strategy; DA+F = data assimilation with the forest strategy

As expected, the simulations assimilating the synthetic SWE observations agreed with the nature run better than the open loop simulation. However, on 13 March 2019 (the date of maximum domain SWE volume), the simulation with data assimilation without the forest strategy had high-biased SWE across large portions of the Rocky Mountains and the Cascade Mountain range (Fig. 1, Fig. 5b and Fig. 5e). Low biased SWE was more common in Northernmost Canadian portions of the Rocky Mountains and Cascade Range, the Western montane regions in Washington State, the Northern portions of the Great Basin, and the lower-lying elevations of the California Sierra Nevada. Additionally, despite the assimilation, snow extents were still biased high relative to the nature run (Fig. 3) at magnitudes similar to the open loop simulation (Table 2). This was driven by the expansive snow extents of the open loop simulation, which were decreased by data assimilation, but still resulted in widespread early-season SWE increases for short periods of time between synthetic observations (at 10 – 14 day

frequencies), increasing to the number of grid cells with DJF SWE exceeding 5 mm (threshold used to define average winter
snow extents in Fig. 3).
Assimilating the synthetic SAR observations without the forest strategy best improved SWE in shrub, grass, crop, bare, and
wetland landcover types (Fig. 6b and 6c). For example, relative to the open loop simulation (Fig. 5a and 5d), data assimilation
without the forest strategy (Fig. 5b and 5e) corrected the high SWE biases in the Great Plains (Fig. 1). While 13 March SWE
in shrub, grass, crop, bare-ground, and wetland regions was typically small in magnitude, these landcover types accounted for
77% of the modeling domain area, and 61% of the domain total SWE volume on 13 March (Fig. 6a). In these regions, SWE
from the open loop simulation had a mean absolute error of 22 mm, and a mean bias of approximately 14%, relative to the
nature run (Table 2). Data assimilation significantly improved the SWE bias in these land cover types to within 1%, on average
(Fig. 6b), with a mean absolute error of 14 mm, relative to the nature run.

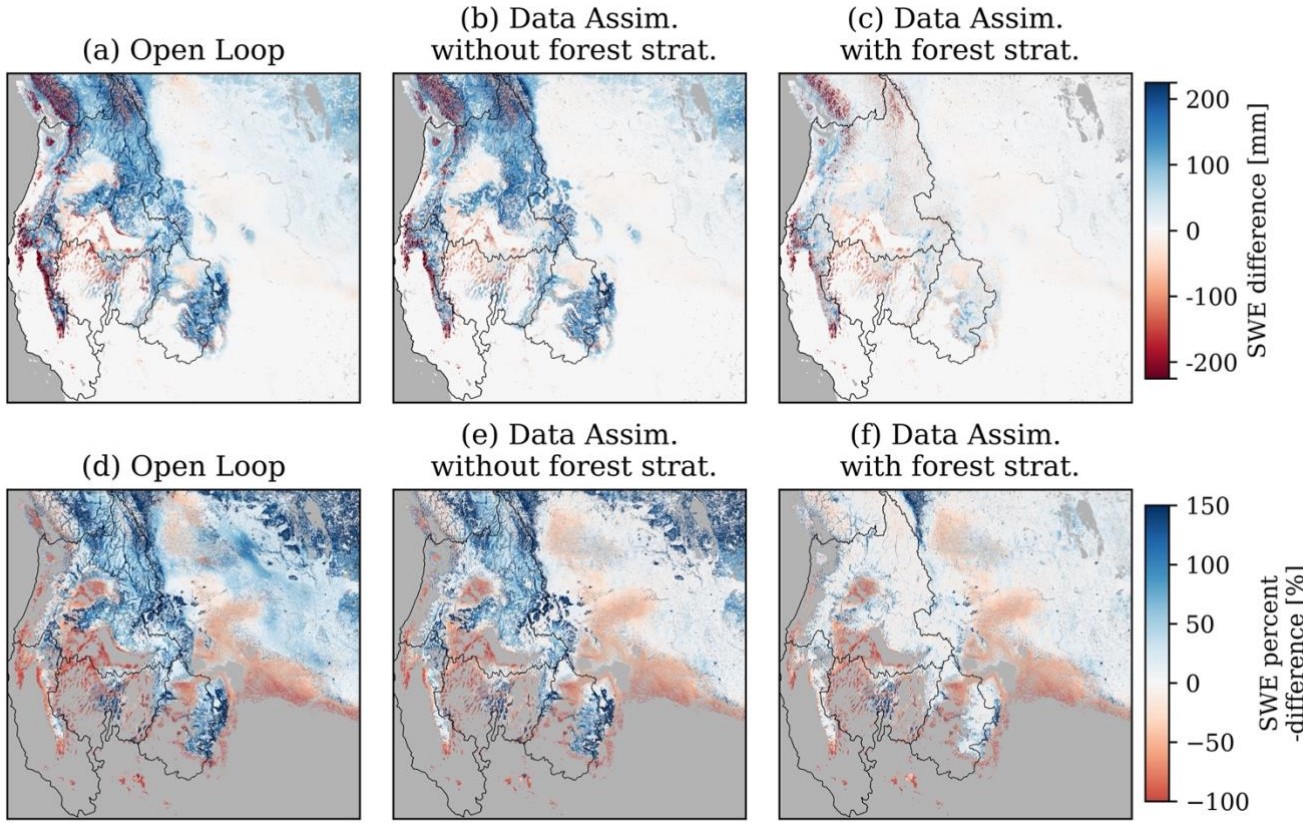

**Figure 5.** 13 March 2019 SWE difference (top row) and percent-difference (bottom row), relative to the nature run, for the
open loop simulation (a and d), and simulations with data assimilation, both with (c and f) and without (b and e) the forest
strategy. SWE percent-different maps (bottom row) only compare grid cells where SWE from the nature run was greater
than 5 mm.
The data assimilation results discussed above did not use the synthetic observations over forested grid cells, where
retrievals from SAR instruments may be either partially or fully occluded by the canopy overstory (Tsang et al., 2022; Ruiz
et al., 2022; Huang et al., 2019). However, a significant portion of the snow volume in mid-latitude domains overlaps with
forests. For example, although forests only covered approximately 22% of the study region investigated here (Fig. 1a),
forested grid cells contained just over 34% of the total 13 March SWE volume, a volume about 10% higher than the snow
volume contained in the next-largest landcover type (Fig. 6a). In forested grid cells, SWE simulated by the open loop
simulation were biased high by approximately 87 mm (+150%) on average (Fig. 6), with a mean absolute error of 111 mm
(Table 2). These errors were propagated into the simulation with data assimilation without the forest strategy. Fortunately,
the ratio between modeled SWE and synthetic SWE observations in forested grid cells and the nearest canopy-free grid cells
had high levels of similarity. Therefore, estimating snow in forest regions using the nearest canopy-free pixels (Fig. 2)
improved snow simulations significantly (Fig. 3d, Fig. 4d, and Fig. 5c and 5f). In fact, snow simulated in forest landscapes
using data assimilation with the forest strategy agreed well with the nature run, exhibiting a 13 March SWE average bias in
forested grid cells of only 14 mm (+8%) (Fig. 6), and a mean absolute error of 27 mm. This forest strategy resulted in large-
scale improvements to total domain SWE (Fig. 5), reducing the 13 March full-domain SWE volume bias by 28%, and
improving the spatial coefficient of correlation by 0.12, relative to the data assimilation simulation without the forest
strategy.

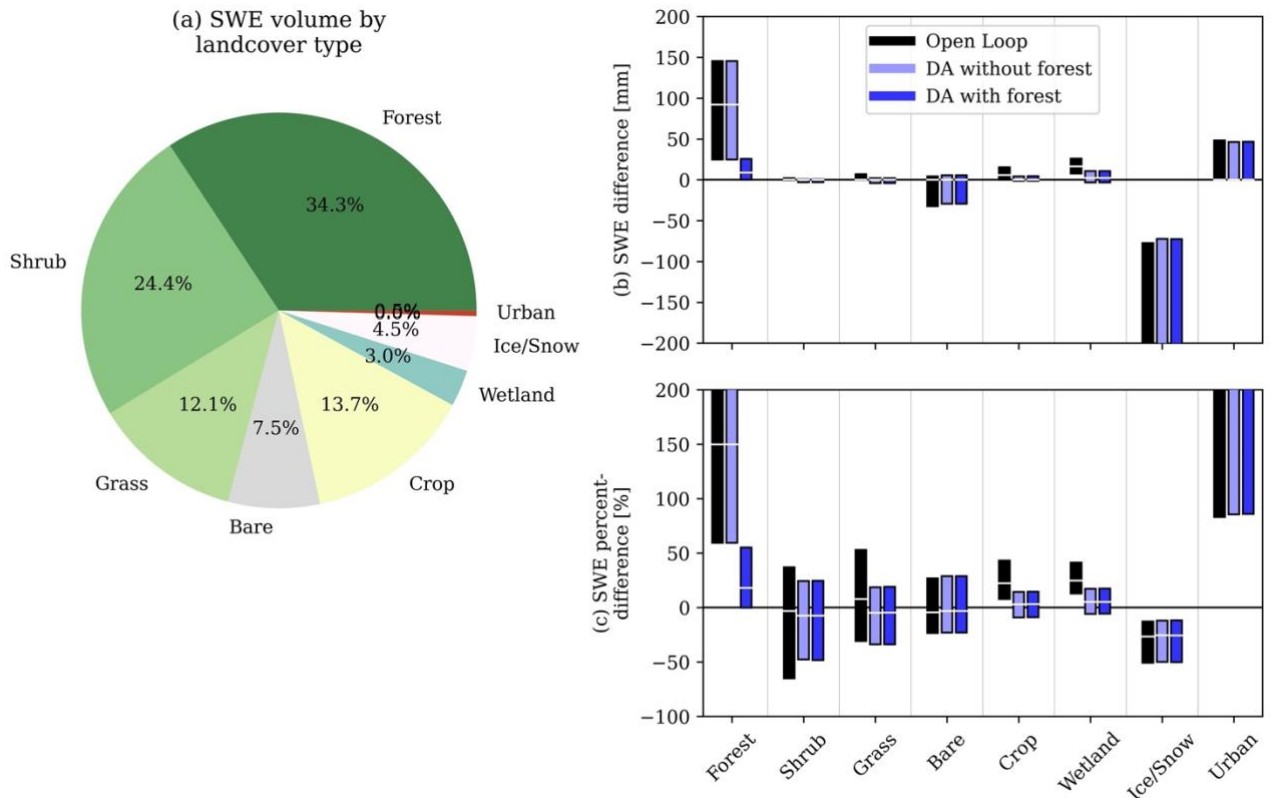


**Figure 6.** SWE volume on 13 March 2019 broken down by landcover type in subplot a. For each landcover type, the interquartile range and median of SWE differences (b) and SWE percent-differences (c) are calculated for the open loop simulation (black) and each simulation with data assimilation (blue bars). SWE differences (b) and (c) are calculated relative to the nature run.

The comparisons above focused on mean DJF SWE and SWE from the date nearest maximum snow volume (13 March, 2019). However, assimilating the synthetic SWE data also improved estimates of snow water resources throughout the duration of the water year, even in periods when most snow-covered regions were experiencing snowmelt and synthetic observations were masked. For example, in the Upper Colorado, approximately 75% of the region had DJF snow cover with little or no winter snowmelt (Fig. 7). The simulation with data assimilation and the forest strategy substantially improved mean SWE evolution in the snow accumulation season in this hydrologic region (Fig. 7, October - March). However, snowmelt onset in the March, April, and May (MAM) months increased the number of grid cells experiencing snowmelt from the open loop model outputs, reducing the number of grid cells across the full Upper Colorado Region that could be observed by the synthetic SAR observations to approximately 5%, on average, over this period of time. Despite this, since the simulation with data assimilation improved the volume, timing, and spatial distribution of maximum SWE, mean SWE evolution tracked the nature run simulation significantly better than the open loop simulation in the spring snowmelt period. In fact, relative to the nature run, MAM SWE from the open loop simulations was biased high by approximately 63%, on

average, in the Upper Colorado (Table 2). The simulation with data assimilation using the forest strategy improved this bias

to less than 1%, on average, over the same period. In this study, simulations using Noah-MP (open loop and data

assimilation simulations) melted snow more rapidly in the later-half of the spring snowmelt season than the nature run

simulation which evolved SWE using SnowModel (Section 2.1). Therefore, although maximum SWE volume, maximum

SWE timing, and MAM SWE were improved by data assimilation, the timing of snow disappearance for the simulation with

data assimilation using the forest strategy was approximately 18 days earlier than the nature run in the Upper Colorado.

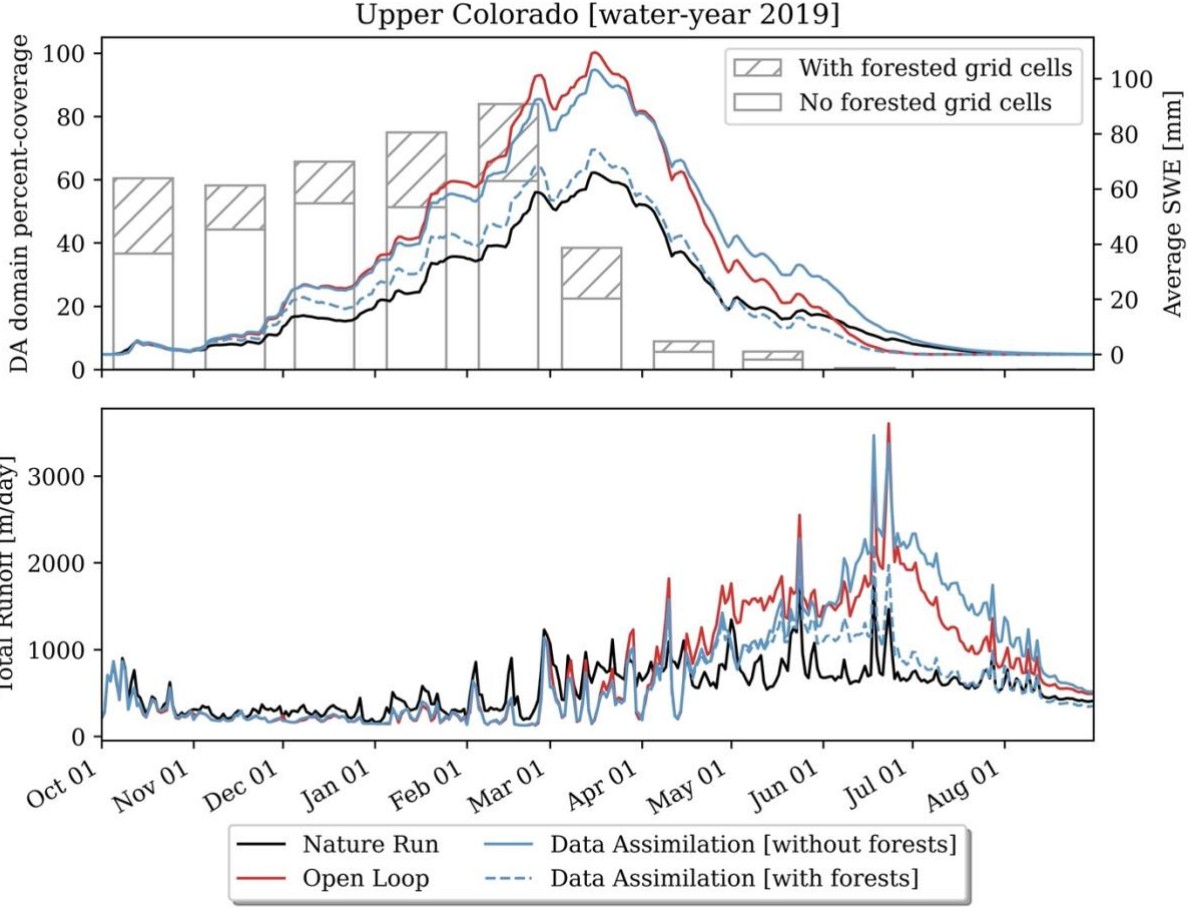

**Figure 7.** Time series comparison of mean SWE (top) and total runoff (bottom) between the open loop, nature run, and
simulations assimilating the synthetic observations, both with and without the forest strategy in the Upper Colorado. Dashed
bars in the top plot represent the monthly percentage of the Upper Colorado grid cells with no snowmelt. Solid bars also
exclude grid cells with forest coverage.

Much like the Upper Colorado, SWE simulated by the open loop simulation in the Pacific Northwest (Fig. S7) was biased

high for the entirety of the snow season. Both domains also had greater than 80% synthetic snow observation coverage in

March (including grid cells that filled snow estimates using the forest strategy), and as a result, the simulation with data

assimilation using the forest strategy closely matched SWE from the nature run. However, both of these domains had a significant portion of the seasonal snowpack in forested landcover (Fig. 7 and Fig. S7, difference between the hatched and solid bars). These grid cells had winter SWE estimates from the open loop simulation that were predominately high-biased (Fig. 3 and Fig. 5). Therefore, although data assimilation improved winter SWE in non-forested landcover types (Fig. 6), the simulation without the forest strategy caused little-to-no improvement in the simulated domain mean maximum SWE (Table 2). This highlights the value of the forest strategy used here (Fig. 2), which drew information from synthetic observations in relatively few nearby pixels to infer the mean snow volume in forested grid cells. Given the four hydrologic regions investigated in this study, a far smaller volume of snow existed in forested landcover for the California (Fig. S8) and Great Basin regions (Fig. S6), resulting in DJF domain-mean SWE evolution that was more similar between the simulations with and without the forest strategy. We expect results in these domains to be more indicative of the value of winter SAR observations in less-vegetated snowy landscapes, such as Tundra and Prairie snow regimes (Sturm and Liston, 2021).

Finally, the improvements to the spatial and temporal estimates of SWE discussed above had trickle-down improvements on simulated runoff. For example, in the Upper Colorado (Fig. 7), total annual runoff from the open loop simulation was biased high by approximately 35%, relative to the nature run. This error was driven most by high-biased winter snow accumulation, which nearly doubled the melt season (March – July) runoff estimated by the nature run simulation. Here, by assimilating the synthetic SWE observations, and estimating forest snowpack from the relationship between modeled and observed SWE from the nearest canopy-free pixels, total annual streamflow in this domain was improved to within 1%. Not only was domain total runoff improved, but the seasonal evolution of high and low-flows vital for water management and planning was also improved. This improved the Nash-Sutcliffe Efficiency (NSE) from –2.59 to 0.22 between the open loop simulation and simulation with data assimilation employing the forest strategy (Table 2). These results were similar for the Pacific Northwest, which had an NSE that improved from –0.17 to 0.39. However, due to the smaller changes to SWE and more-rapid snowmelt simulated by Noah-MP, changes to runoff from data assimilation in California and the Great Basin were small (Table 2), with improvements that were largely outweighed by the difference in snowmelt timing and rates between Noah-MP and SnowModel.

**4. Discussion**

The differences between the open loop simulation and nature run in this study were representative of snow modeling errors common for continental and global-scale models used for seasonal to long-term future snow predictions (e.g., Franz et al., 2010; Garousi-Nejad and Tarboton, 2022; Kim et al, 2021; Liu et al 2022). The greatest source of these snow modeling errors is commonly errors in meteorological forcing data, and in particular, biases in precipitation (Garousi-Nejad and Tarboton, 2022; Henn et al., 2018; Pflug et al., 2021; Raleigh and Lundquist, 2012; Wayand et al., 2013). These biases are especially prevalent in the portions of the earth's surface with the greatest volumes of snow, such as the tundra and montane regions (Kim et al., 2021), where ground observations and observation station maintenance are hindered by harsh winter conditions and

inaccessibility. This suggests that the greatest need for improving global estimates of snow is improved estimates of snow accumulation in remote, under sampled landscapes. Here, we expect that the SAR observations evaluated in this study could address these needs, thus providing a path forward for pairing common snow models with observations as a basis for determining global snow mass. For example, assimilating SAR observations at 10 – 14 day intervals with the observational error characteristics reported in Sect. 2.2, improved midlatitude winter SWE volume by approximately 22%, on average (Table 2). In unforested landscapes, which account for a majority of the Earth's snow water storage (Kim et al., 2021), assimilation improved the mean SWE bias at maximum SWE timing to within 1%, on average, and reduced the standard deviation of errors by approximately 45 mm (~85%) (Fig. 6).

Despite the benefits discussed above, SAR observations have known limitations in forested landscapes where the canopy overstory obstructs retrievals from the underlying snowpack (Huang et al., 2019; Ruiz et al., 2022; Tsang et al., 2022). Therefore, this study was designed to investigate a forest strategy that uses the relationship between modeled SWE estimates and synthetic SWE observations from neighboring grid cells as the basis for inferring snow distribution in regions with forested landcover (Fig. 2). To focus on the benefits of this approach, we chose a domain (Fig. 1) that included both significant forest spatial coverage (22%) with disproportionate amounts of winter snow (34%) within the forested pixels (Fig. 6). Relative to the open loop simulation, the simulation with data assimilation and the forest strategy dramatically improved the spatial distribution of SWE (e.g., Fig. 3 and Fig. 4) and the resulting SWE biases at domain maximum snowpack timing (Fig. 5). In fact, in forested grid cells, SWE on 13 March was only biased by 14 mm (mean absolute error of 27 mm), on average, for the simulation with data assimilation and the forest strategy, relative to the nature run. This was opposed to the open loop simulation, which was biased by 87 mm (mean absolute error of 111 mm) over the same regions and date. Despite the fact that the two simulations with data assimilation agreed in all grid cells except forested grid cells, the simulation employing the forest strategy had a mean absolute error (17 mm) across the full modeling region that was approximately 51% smaller than the simulation without the forest strategy. Here, we recognize that this study used a single date (13 March) to represent snow water resources at maximum SWE timing. However, the date of maximum SWE volume from the nature run varied by less than a week across the four hydrologic regions (11 - 16 March; Fig. 7, Fig. S6 - S8). Therefore, this was a relevant date for model comparisons, especially given that water resource and allocation decisions in the Western US are often based on the volume of snow at maximum snow timing.

This research shows how a modeling framework and relatively few observations can be used to gap-fill estimates of snow in regions where remote sensing observations from a future platform may be most challenged. Despite the fact that snowpack with properties able to be retrieved by SAR instrumentation (i.e., canopy-free landcover and no snowmelt) sometimes only accounted for only small portions of a modeling domain (e.g., Fig. 7), SWE from the model and SAR observations in nearby canopy-free grid cells were predictive of the snow in forested grid cells. We hypothesize that this could have partly been driven by the 250 m resolution of synthetic observations and simulations. At this length scale, snow distribution is typically driven

by processes like mesoscale weather patterns and their interaction (e.g., orographic lapse rates, wind loading/sheltering, terrain-shading, etc.) with static topographical features like elevation, slope, and aspect (e.g., Clark et al., 2011; Lehning et al., 2011; McGrath et al., 2018; Minder et al., 2008; Trujillo et al., 2007). However, we acknowledge that snow in forested and open grid cells is subject to different snow processes. In fact, the nature run simulation used here attempts to simulate snow-canopy interactions, such as snow interception and solar shading from the canopy overstory (Liston and Elder, 2006). Here, since we focus predominantly on model improvements from data assimilation in the SWE accumulation season, we hypothesize that the primary difference between SWE accumulation in forested pixels, and the nearest canopy-free grid cells could be driven by canopy interception, or the lack thereof. In other words, inferring forested snowpack using the nearest canopy-free grid cells could bias snow in forested regions where snow processes differ slightly. While the forest strategy improved SWE simulated in forested grid cells at the date of maximum SWE volume, SWE was still biased high relative to the nature run (Fig. 6). We hypothesize that a correction factor, based on variables like forest canopy type, vegetation density, wind speed, and temperature during snowfall, all of which influence snow interception (Lundquist et al., 2021), could be used to facilitate the difference in snow accumulation expected between a forest pixel and SWE observations from nearby canopy-free grid cells. This approach will be a topic of future research. However, since errors with precipitation are often the overwhelming source of model errors, we hypothesize that the forest strategy (Fig. 2), which corrected modeled SWE in forested areas using the ratio between modeled and observed SWE in nearby open areas, was well-suited to correct precipitation biases.

The results presented here are subject to a number of assumptions. These assumptions were intended to apply regionally-consistent and conservative rules about how 1) synthetic SAR observations were generated, and 2) the grid cells and time periods that SAR observations occurred in. For example, we used a 20% and zero-mean random distribution of errors to generate observations from the nature run. We expect the error from a future satellite mission to be less than 20% over the majority of snow covered regions (Sect. 2.2). However, observational biases may be more common in certain locations and periods based on snow depth, particularly in very shallow or very deep snowpacks, terrain characteristics and vegetation characteristics. Additionally, the landcover classification used in this study (Fig. 1) was based on the dominant landcover type within each model grid cell, as defined from the North American Land Change Monitoring System (Latifovic et al., 2017). For forested grid cells, this included needleleaf, broadleaf, and mixed forest types. To be conservative, this study completely masked synthetic observations in 250 m grid cells classified as forest, thereby assuming 1) no observation capabilities in predominantly forested areas, and 2) full observation capabilities in grid cells where forests were not the dominant landcover type. In reality, SAR may be able to achieve accurate snow retrievals in some forested-dominated regions based on the forest type, forest distribution, and canopy density (Tsang et al., 2022). Conversely, some regions with sparser or no forest cover may still have observation limitations based on the domain and snow characteristics mentioned above. The large domain used in this study also made tests over multiple years computationally challenging. Here, the intent of this study was to investigate a strategy for deriving SWE corrections in difficult to observe forest landscapes, and we hypothesize that precipitation biases and the resulting modeled SWE accumulation could be improved to a similar degree in years with both larger and smaller

snow volumes. Finally, while strategies for identifying and correcting systematic SAR observation errors are a topic of
continued research (e.g., Durand et al., 2023; Singh et al., 2023), OSSEs are an inherently flexible framework for evaluating
sensor utility, so future research could use the simulations performed here to test a wider array of sensor configurations and
non-normal retrieval errors. Future work could build upon these results to investigate multiple years, perhaps considering
warmer and/or drier snow years, when the role of snowpack for water supply and midwinter snowmelt and rain-on-snow
frequency may be more likely to increase snowpack liquid water content, or years with late-season spring snow accumulation.
Future research should also investigate other gap-filling approaches, like methods to infer SWE in grid cells where snowmelt
is occurring and liquid water may prevent SAR retrievals, and gap-filling approaches using different window sizes and/or
searching windows that more heavily weight unforested grid cells with similar characteristics (elevation, aspect, etc.).
This study tested a simple model setup using a popular land surface model (Noah-MP) and Kalman-based data assimilation
procedure. This data assimilation procedure updated modeled snow states, like snow depth and SWE, based only on synthetic
SWE observations at 10 – 14 day temporal frequencies where/when snowmelt was not occurring. Despite the limitations and
assumptions discussed above, we expect that the results presented here could represent the lower-bound of performance that
could be achieved from a real-time modeling framework that could accompany a space-borne SAR remote sensing platform.
For example, many studies have demonstrated repeatable patterns of snow accumulation in years with similar winter
meteorological characteristics (e.g., Deems et al., 2008; Pflug et al, 2022; Schirmer et al., 2011; Sturm and Wagner, 2010;
Woodruff and Qualls, 2019). This suggests that retrospective information about snow distribution patterns in previous years,
could be used as the basis for extrapolating and updating snow model states in grid cells not covered by SAR observations on
a given date. From the modeling perspective, only 5 ensemble members were used in the Ensemble-Kalman data assimilation
(Sec 2.3), when a larger ensemble of simulations may have improved uncertainty characterization of simulated snow and
hydrological states even more. This study also assumed that synthetic SAR observations were unable to observe snow in all
forested landscapes, when retrievals of snow in forested stands could be achievable for some forested regions with smaller tree
cover fractions and biomass (Montomoli et al., 2015; Tsang et al., 2022). Finally, the SAR configuration tested here had 10 –
14 day repeat times, but future satellite configurations with more-frequent observational repeats are possible and have been
recommended by the 2018 Decadal Survey (NASEM 2018). Despite all of these conservative assumptions, the difference
between the open loop simulation (representative of current modeling accuracies), and the simulation with synthetic
observation data assimilation using the forest strategy, demonstrated large-magnitude and widespread improvements to real-
time estimates of winter SWE and the associated improvement to spring SWE and runoff. Therefore, we expect the findings
of this study, particularly the strategy to extend the observational utility to forested areas, to significantly aid in the full
exploitation of the information from a future SAR-based snow satellite mission.
5. **Conclusion**
Global estimates of snow volume and distribution have uncertainties stemming from limited snow observations and
biases in meteorological forcing data. These uncertainties stress the need for a global snow-focused satellite remote
sensing platform. Here, we investigate the degree to which synthetic observations of SWE representative of a Synthetic
Aperture Radar (SAR) remote sensing platform, could correct common snow modeling errors and provide
spatiotemporally continuous SWE estimates. We investigate this using an Observing System Simulation Experiment,
specifically investigating how much snow simulated using a widely used land surface model and meteorological forcing
dataset, could be improved by assimilating synthetic SAR observations of SWE.
The difference between the open loop simulation and the nature run was representative of common modeling errors.
Snow simulated by the open loop simulation had larger winter snow extents, and total snow volume that was biased high
by approximately 35%. The open loop simulation also simulated snow that was more spatially homogeneous,
underestimating the variability across variations in topography and underestimating lower-elevation snowmelt from the
nature run. Assimilating the synthetic SWE observations improved SWE simulated in the shrub, grass, crop, bare-ground,
and wetland land cover types. In fact, SWE biases on the date of domain maximum SWE volume (13 March 2019) in
these landcover types improved from 14% for the open loop simulation to within 1% after data assimilation. However,
despite only covering 22% of the study area, forested grid cells contained just over 34% of the domain SWE on 13
March. The open loop simulation and the simulation with data assimilation without the forest strategy had SWE that was
high biased by 150% (87 mm), on average, in these forested grid cells.  The relationship between modeled SWE and
synthetic SWE observations in forested grid cells exhibited similarities with the nearest canopy-free grid cells. Therefore,
SWE in forested regions was able to be inferred using the simple modeling framework and synthetic SAR observations
from nearby canopy-free grid cells. In fact, the simulation with data assimilation using this forest gap-filling strategy
substantially improved SWE biases to 4% (~22% improvement) at maximum SWE timing, with a SWE mean absolute
error of 17 mm (24 mm improvement) and spatial correlation of 0.91 (0.17 improvement) across the Western US
Improvements in winter SWE accumulation also improved estimates of melt-season SWE evolution and total runoff
in four major Western US hydrologic regions, even in periods when winter snowmelt greatly reduced the number of grid
cells that could be observed by the synthetic SWE observations. In fact, in the Upper Colorado River, melt season SWE
biases improved from 63% to less than 1% after assimilation, and the runoff Nash Sutcliffe Efficiency improved from -
2.59 to 0.22. These results demonstrate the value of SAR observations and simple spatial-filling strategies in grid cells
where SAR retrievals could be obstructed by the canopy. Here, we expect our results to represent a lower-boundary of
model performance which could be improved further by more robust assimilation approaches, more-frequent SAR
observations, further developments to SAR retrieval algorithms in forested landscapes, and adaptations to the forest gap-
filling strategy developed here. However, our results also show that widespread improvements to global SWE could be
available in near real-time provided data assimilation approaches and a SAR remote sensing platform.
*Code availability:* The Land Information System (LIS; lis.gsfc.nasa.gov) framework used to perform the nature run, open
loop, and data assimilation simulations from this study can be accessed from a GitHub public repository
(https://github.com/NASA-LIS/LISF). Model documentation and LIS tutorials can also be accessed from this repository. Users
are encouraged to reference Kumar et al. (2006) for more information on LIS. The Trade-space Analysis Tool for designing
Constellations (TAT-C) tool is currently available on-request for federal employees and contractors
(https://software.nasa.gov/software/GSC-18399-1).
*Data availability:* The model outputs and data necessary to reproduce the figures and statistics reported in this study can
be found at https://www.hydroshare.org/resource/e0ad80f818bf4062a335e9e0d7362834/. This repository includes
domain static variables, such as land cover, elevation, and spatial coordinates, in addition to model outputs of winter-average
SWE, SWE at the date of maximum SWE volume (13 March 2019), and SWE and runoff aggregated across each region.
MERRA-2 forcing data can be accessed from the Goddard Earth Sciences Data and Information Services Center (GES DISC,
https://disc.gsfc.nasa.gov/), and ERA5 data can be accessed from the European Centre for Medium-Range Weather Forecasts
climate data store (https://www.ecmwf.int/en/forecasts/dataset/ecmwf-reanalysis-v5).
*Author contributions:* CV and SK coordinated the manuscript question and research methodology. SK adapted LIS model
source code to implement the forest gap-filling strategy in Sect. 2.4. MW set up the model domains and model configurations,
and performed the open loop simulation. EC assisted with generating the synthetic SWE observations. With assistance from
KA, JP performed code developments for the simulations using both the Noah-MP and coupled SnowModel models. KA
implemented the SnowModel code and parameters into LIS and LDT. JP also performed the nature run and both data
assimilation simulations. The manuscript was written provided text, figures, and feedback from all coauthors.
*Competing interests:* The authors declare that they have no conflict of interest.
*Acknowledgements:* This work was supported by the funding from the NASA Terrestrial Hydrology program. Computing was
supported by the NASA Center for Climate Simulation (NCCS). We would also like to acknowledge Dr. Ethan Gutmann and
NASA Grant 80NSSC20K0207 for supporting the integration of the SnowModel distributed model into LIS.

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
