# Peer review of "Extending the utility of space-borne snow water equivalent"

_EGUsphere, 2023_

## Author Response (AR1)

**Response to RC1**

We would like to thank anonymous reviewer #1 for their thorough and thoughtful comments. We believe that by addressing these comments we greatly improved the clarity and organization of the manuscript. Below, we include in-line responses for each of the reviewer's comments.

**Detailed comments:**

1) *The descriptions of "Open loop", "Nature Run", "Assimilation w/o Forest" and assimilation with forest are scattered and hard to follow. Readers not familiar with the concept of open loop and nature run will struggle and it will be helpful to provide a summary. I am unclear on what biases and uncertainties were considered in the open loop simulations*

   Thanks for pointing this out. We agree that our terminology for these simulations was sometimes inconsistent and confusing. We changed the text throughout the manuscript to make our discussion of the different simulations and labeling for each clearer. For example, additional text was added to link specific simulations in the text with the labels used in the figures (lines 248 - 249 and lines 268 - 269).

   We also agree that the differences between the open loop and nature run simulations were not stated as clearly as they could have been. The differences between these simulations were driven by the different models simulating snow (Noah-MP versus SnowModel), model forcing (ERA5 versus MERRA2), and different meteorological downscaling procedures. We edited Section 2.1 to ensure that the nature run and open loop simulations each have a paragraph dedicated to discussing their setups, and added a table to the supplementary to further highlight the differences (Table S1). We also note that the disparity between the open loop and nature run simulations is representative of snow modeling errors that are common for continental and global scale models (lines 414 - 416). The improvement to the simulated SWE following assimilation shows that a snow satellite with the characteristics presented here, paired with the forest strategy for adjusting snow in difficult-to-observe locations, could correct these common regional/global model errors (lines 414 - 428).

2) *How did you come up with the 750 m search radius for the forested and non-forested pairing?*

   This 750 m searching radius corresponded to a 5x5 window of grid cells. Given the large expanse of forests in this domain (Fig. 1), smaller searching radii increased the prevalence of windows where every cell within the window was forested and thereby unable to infer the SWE correction. Secondly, the forest strategy worked well due to the spatial similarity between the winter SWE corrections in forested grid cells and the SWE corrections from the nearest unforested grid cells. We attributed this to the high spatial autocorrelation in the precipitation differences between the open loop and nature run simulations (lines 451 - 455). Given this, we expected that a larger searching radius would increase the risk of including SWE corrections at grid cells with dissimilar SWE errors. In lines 493 - 495, we acknowledge that future work could investigate different

searching approaches, including different window sizes or gap-filling approaches that more heavily weight grid cells with similar topographical features.

*3) Considering the fact that SWE magnitude and processes are highly variable among different years, how transferable are results from 2019 to a dry year when the role of snowpack is even greater for managing water? This point warrants a discussion.*

This is a great point. In this manuscript, we chose to focus our efforts and results on the forest strategy (Section 2.4), which addressed one of the greatest challenges for SAR remote sensing retrievals. Due to this and the large computational cost of these simulations (lines 134 - 138), we chose to focus the analysis on only a single water year. Based on our results, we anticipate that the improvements that we presented here would likely be similar for both higher and lower snow years. Specifically, we expect that the results can be transferable to dry years provided that 1) the distribution of forested/non-forested snow (e.g., Figure 6) was similar, and 2) the disparity between the modeled snow biases in forested and nearby unforested grid cells was also similar. However, we agree that this should be evaluated in future research. We discuss and acknowledge this in lines 487 - 493.

*4) You are using snow water equivalent and SWE interchangeably. use the abbreviation after you have defined it.*

Thanks for pointing this out. These instances have been corrected (e.g., line 53).

*5) L58 missing reference*

Thanks for catching this. These citations were added (line 59).

*6) Fig. 1: Red contours are not really basins. They are boundaries of water resources regions 14 (upper Colorado), 16 (Great Basin), 17 (PNW), and 18 (CA).*

Good point. We removed our use of "basins" in the text (e.g., line 131). We now use the term "hydrologic region" throughout.

*7) L143: use of Western United States vs Western U.S.*

All instances except the first instance of "Western United States", were changed to "Western US" for consistency.

*8) Is it fair to use the same peak snow day, i.e., March 13 2009, for the entire study region? How variable this peak date was among the four regions?*

This is a great question. This date was selected based on the date of maximum SWE volume across the full modeling domain. However, the date of maximum SWE volume from the nature run only varied between March 11 and 16 across the four hydrologic regions (see Figure 7, and Figures S6 – S8). Therefore, we believe that this is a

reasonable target date for model evaluation, especially given that water resource allocation and management decisions in the Western US are often based on the volume of snow at maximum snow timing (lines 442 - 446).

Although maximum snow volume peaked around 13 March, many lower-elevation grid cells were already experiencing melt by that date, causing snow cover to have high liquid water content and eliminating the capability of SAR observations on that date (e.g., bars in Figure 7). Therefore, many of the SWE errors reported in Table 1, Figure 5, and Figure 6 are driven by drift between the open loop and nature run, which used different models, meteorological forcing, and downscaling. In other words, the model performance that we report for 13 March was worse than if we had focused on an earlier winter date when less snowmelt was occurring and greater amounts of simulated observations were included in the assimilation.

**Response to RC2**

We would like to thank anonymous reviewer #2 for their comments and suggestions. We believe that this manuscript greatly benefitted from addressing these comments. Below, we include detailed responses for each of the reviewer's comments.

**Detailed comments:**

1. *The way it is written it appears the "nature run" is used as the ground truth. At the large spatial extent of the Western US this seems appropriate. However, it is unknown how the "nature run" compares to ground based SWE observations. It would be helpful to provide some comparisons to observed SWE from SNOTEL stations in different regions to show that the whole modeling framework is actually representative of actual "ground truth".*

   Thanks for the comment. In the manuscript, we followed the approach typical among OSSE studies (e.g., Errico et al., 2007; Garnaud et al., 2019; Kwon et al., 2021) by focusing on developing an experiment setup that uses the departure between the open loop and the nature run to simulate errors common for continental and global snow simulations (lines 414 - 416). Overall, our setup allowed us to test the degree of model adjustment and resulting SWE estimation that could be provided from a satellite snow sensor with the characteristics provided here (Sect. 2.4). Given the size of our domain and spatial resolution, model calibration and bias corrections to the meteorological forcing data would be time and computationally expensive, and would likely require different model calibrations and corrections based on regionally and temporally specific errors. Therefore, we relied on the use of a state-of-the-art model with improved precipitation forcing and downscaling techniques (SnowModel coupled with Noah-MP, NASA's MERRA-2 forcing, and slope/aspect-based meteorological downscaling) to represent the "true" state of snow evolution, and a more-common modeling setup (Noah-MP and ERA5 forcing) to represent the open loop model states.

   Though ensuring the realism of the nature run is not the primary focus of OSSEs, we established that snow evolution from the nature run resembled realistic conditions in the Western US and Canada. We added some comparisons between our simulations and a widely-used Western US snow reanalysis (Fang et al., 2022) which has been validated with ground-based SWE observations from SNOTEL locations. We added this comparison to the supplemental material and referenced it in the main text (lines 175 - 182). We chose to compare our simulations with this reanalysis product since comparisons between grid cells (250 m) and point observations of SWE can be challenging. We found that the extent to which the nature run matched SWE from the reanalysis in 2019 varied regionally. For instance, on 13 March (date used to approximate peak snowpack timing in this study), SWE from the nature run and reanalysis had a spatial coefficient of correlation between 0.31 and 0.42 in a Pacific Northwest and Colorado subregion. This was significantly better than the open loop simulation, which only had a coefficient of correlation of approximately 0.15. Further, snow cover from the nature run was biased by +4% relative to the reanalysis, but the open loop consistently overestimated snow cover by greater than +13%. While total snow volume between the

nature run and reanalysis agreed closely in the Pacific Northwest subdomain, SWE from the nature run was biased low in the Colorado subdomain.

2. *Can you comment on why on a single year (2019) was used? I realize the modeling is likely a big lift computationally but snowpack characteristics can change an awful lot from year to year. You picked a big snowpack year but what about a shallow snowpack year?*

   This is a great question. In this manuscript, we chose to focus our efforts and results on the forest strategy (Section 2.4), which addressed one of the greatest challenges for SAR remote sensing retrievals. Due to this, and due to the large computational cost of these simulations (lines 134 - 138), we chose to focus the analysis on only a single water year. Based on our results, we anticipate that the improvements that we presented here would likely be similar for both higher and lower snow years. However, this could be evaluated in future research. We now discuss and acknowledge this in lines 487 - 493. We also addressed a similar comment from Reviewer #1 (question #3).

3. *The introduction mentions a "future snow mission". Can you clarify if this is an already planned mission or more of a hypothetical mission?*

   The "future snow mission" referenced in this manuscript is a hypothetical mission at this time. However, much of the snow community, including the coauthors on this study, and scientists from both the US and international communities, have been researching what the capabilities of a snow-focused remote sensing platform could look like (e.g., Cho et al., 2023; Garnaud et al., 2019) in preparation for satellite mission opportunities. This manuscript, which demonstrates the value of SAR-based snow observations combined with modeling efforts, contributes to that future mission effort.

4. *In the Short Summary it says "250 m estimates of snow". Can you be more specific and say snow water equivalent. It also states "snow water volume to within 4%". Can you add what that is in reference to? Ground truth?*

   Good idea. This was changed to "snow water equivalent". The short summary is restricted in length (500 characters, including spaces), so I was unable to add that the reference for this statistic was the nature run. However, I added text to the abstract to clarify this.

5. *Line 58: It looks like "Ref" needs to be filled in.*

   We apologize for the oversight and thanks for catching this. This was edited (line 59)

6. *Line 131: What does "SW" mean? Please define.*

   Thanks for pointing this out. This was supposed to be "shortwave". This was changed in the updated manuscript.

7. *Line 245: Is the word "snowy" out of place here?*

Good catch. This was a typo, and it was removed in the updated text.

References:

Cho, E., Vuyovich, C. M., Kumar, S. V., Wrzesien, M. L., & Kim, R. S. (2023). Evaluating the utility of active microwave observations as a snow mission concept using observing system simulation experiments. The Cryosphere, 17(9), 3915-3931.
Errico, R.M., Yang, R., Masutani, M., Woollen, J.S.: The estimation of analysis error characteristics using an observation systems simulation experiment. Meteorologische Zeitschrift 16, 695–708, 2007.
Garnaud, C., Bélair, S., Carrera, M.L., Derksen, C., Bilodeau, B., Abrahamowicz, M., Gauthier, N., Vionnet, V.: Quantifying Snow Mass Mission Concept Trade-Offs Using an Observing System Simulation Experiment. Journal of Hydrometeorology 20, 155–173. https://doi.org/10.1175/JHM-D-17-0241.1, 2019.
Kwon, Y., Yoon, Y., Forman, B.A., Kumar, S.V., Wang, L.: Quantifying the observational requirements of a space-borne LiDAR snow mission. Journal of Hydrology 601, 126709. https://doi.org/10.1016/j.jhydrol.2021.126709, 2021.

**Response to RC3**

*I have a number of questions below, focused on the experimental setup. In short, some additional details on: the radar mission configuration, sensitivity of the results to the 20% uncertainty value prescribed to the synthetic SWE data, and some assurances about the independence of the results from the nature run itself are needed before the manuscript is suitable for publication.*

We would like to thank reviewer #3 for their thorough review and thoughtful comments. It was clear to us that this reviewer is familiar with the SAR remote sensing, and we believe that their comments on this and the related study design greatly improved our manuscript. We also agree that the points that reviewer #3 raised here and detailed later on should have been addressed to greater detail in the original manuscript. Below, we include more detailed responses to each of these overarching concerns, including references to where content was added or changed in the revised manuscript.

In summary, the radar configuration used in this study wasn't selected to correspond directly with any proposed mission, but was instead intended to represent a conservative retrievals that are likely worse than we would expect from a future snow-focused satellite. We added text to emphasize this, and we specifically address the reviewer's comments on 1) retrieval uncertainties, 2) uncertainties stemming from heterogeneities in the land cover, and 3) systematic biases that could result from land/snow characteristics and viewing geometries. Additionally, in the revised manuscript, we emphasize that the setup used here is a "fraternal twin" OSSE where we use different models (Noah-MP and SnowModel) to simulate snow evolution. Relative to "indentical twin" OSSEs, which use the same model, "fraternal twin" OSSEs have greater independence between the model states and information content which better avoids the internal consistencies between the open loop and nature run (e.g., Yu et al., 2019) that the reviewer is expressing concerns about.

More detailed responses to these concerns are included in the detailed comments below. Again, we would like to thank reviewer #3 for their time and thoughtful comments. We feel well served by this review, and look forward to their feedback on our responses and edits.

**Detailed comments:**

1. *Section 2.2: It would be helpful to have some further details on the exact mission configuration, or range of configurations that motivated this study would be helpful. There is only a parenthetical statement on line 83 that the focus is a volume-scattering based approach using X- and Ku-band measurements. Were a number of mission and orbital configurations explored? It's not clear what swath width, or range of swath widths were applied and it is only stated that a 10 to 14 day revisit time was used.*

    The focus of this study was to demonstrate the value of the forest gap-filling strategy. Therefore, we are not showing results from a range of different mission configurations here. Instead we chose a conservative configuration, meaning one with less capability

than we anticipate from a future snow mission, to show that even with fewer observations and worse-than-expected accuracies we expect improved SWE estimates, and by employing a strategy over forested regions those estimates can be improved further still. We assimilated observations every 10-14 days, at a 250 m resolution which is representative of a small constellation of SAR satellites. We revised Section 2.2 to better highlight this focus.

2. *Line 166: "Additionally, based on an error level of 20%, spatially and temporally uncorrelated random errors drawn from a Gaussian distribution were added to the synthetic SWE retrievals." Does this mean the 20% random error was applied directly to the nature run SWE values? While 20% isn't overly conservative, there are systematic error considerations based on the frequency of radar measurement, and radar geometry considerations that result from the configuration of the mission. For instance, Ku-band retrievals will be biased low in deep snow areas while X-band will be more insensitive to shallow snow. Measurements near swath edges may have an incidence angle that results in systematic errors. Some additional details on how this 20% number was determined, whether you explored the sensitivity of the results to this value, and the potential impact of not considering more mission-specific systematic errors would be helpful.*

   Thank you for this comment. We agree that the error assumption was not well described and required further explanation. Yes, the random 20% gaussian error was uniformly applied to the portions of the nature run that fell within orbital "cookie cutter" swaths on each day. This is a conservative estimate of the expected error from a combined X-/Ku-band SAR system, based on previous mission design concept efforts. For example, the CoREH2O mission expected to meet requirements of +/-30 mm for SWE  300 mm and +/-10% for SWE > 300 mm (Rott et al 2010, 2012) which accounted for system and instrument error, including swath edge effects, as well as SWE retrieval performance. Similarly, the Canadian Terrestrial Snow Mass Mission (TSMM) concept that is currently under development aims to achieve better than 20% measurement uncertainty for SWE greater than 50 mm, though it is limited to SWE less than 200 mm (Garnaud et al. 2019). Combined X- and Ku-band SWE retrievals have been successfully demonstrated over a range of snow conditions based on recent airborne and tower-based validation efforts (Zhu et al. 2018, 2021, Tsang et al 2022, Durand et al. 2023, Singh et al. 2023). While we expect a future snow mission to achieve better than 20% measurement uncertainty over the majority of snow covered area, we applied this conservative error estimate uniformly over the observational swath.

   Given this manuscript's primary focus on the forest-filling strategy (Section 2.4) and computationally-expensive model domain (~83 million grid cells and 1000 processors), we wanted to keep our narrative and methods simple, simulating results for a configuration that could be worse than whatever may actually be operationally-feasible. However, we also agree that observational biases may be more common in certain locations and periods, particularly in very shallow or very deep snowpacks. We acknowledge this at the end of Section 2.2, and we added an additional discussion on this in the second-to-last paragraph of Section 4. We point out that while the size of these domains made simulations with multiple satellite observation characteristics/accuracies

computationally-expensive, we expect the simulations performed here to serve as the basis for future studies, which could use subsets of the results from this study to investigate smaller regions where certain retrieval errors may be more prevalent and/or systematic (lines 483 - 490).

3. *Line 211 - 219: the approach to filling in forested areas with information from adjacent non-forested grid cells is clearly described and nicely illustrated in Figure 2. The text and image generally gives the impression that gridded SAR backscatter will either be from a clearing or a forest. But vegetation cover relevant to radar backscatter is not a binary influence. What thresholds were used to differentiate forest from clearing? How would mixed grid cells be treated? What about the influence of non-forest vegetation like shrubs within the snowpack?*

These questions raise some good points. We recognize that SAR retrievals would be influenced by landcover heterogeneities, which could include multiple forest densities, distributions, and forest types. Here, our approach was conservative by prescribing grid cells as fully "forested" and thereby unobservable if the most-dominant land cover for any gridcell was forested. However, you are correct that forested/unforested landcover is not binary in reality, and that some grid cells with more forest cover could still have accurate SAR retrievals while nearby areas with sparser forest or other vegetation types and terrain features could be subject to inaccuracies.

Given the size of the modeling domain investigated here, we believe that our land classification approach was the most strict and spatially consistent way to partition grid cells that were the most/least likely to be obscured by the forest canopy, on average. However, the degree to which different arrangements, densities, and forest types influence SAR retrievals is a topic of continued research (including research from coauthors on this study). We also added an acknowledgement of this study assumption in lines 476 - 483.

4. *Line 230: "when snowmelt is minimized and synthetic observations are masked by grid cells with liquid water content to the smallest degree". This text suggests that wet snow grid cells were not assimilated, similar to how forest cells were masked. If this is the case, a description of how wet snow was treated needs to be added to Section 2.3 or 2.4.*

This is a great suggestion. As previously-written, the fact that melting-snow grid cells were masked out in the observation simulator was briefly mentioned at the end of Section 2.3, but should have been highlighted more. We added an additional sentence to clarify which grid cells were included in the assimilation (lines 248 - 249 and lines 268 - 269). We also revised the first paragraph of Section 3 to remind readers of the differences between the simulations and which grid cells were included in the data assimilation procedure (Lines 281 - 284). Finally, we added a supplemental table to summarize the simulation differences (Table S1).

5. *Line 264: "This was driven by the expansive snow extents of the open loop simulation…". A feature of the open loop simulation is the smoother spatial pattern and clear lack of elevational influence on SWE. The high SWE areas are very smooth, unlike the nature run. In*

*trying to understand this, I went back to Section 2, but could not find a clear description of the open loop simulation. Based on line 134, it is stated that Noah-MP (without SnowModel) was used for the open loop simulation but some further insight into the underlying differences in Figures 3 and 4 would be helpful.*

Thanks for pointing this out. This was something that was also noted by the other reviewers. We revised Section 2 and added a table to the supplementary (Table S1) to more clearly indicate the differences between the simulations. In summary, the differences were a result of different snow models, model forcing, and meteorological downscaling procedures.

6. *Line 290: "In forested grid cells, SWE simulated by the open loop simulation were biased high by approximately 87 mm (+150%) on average (Fig. 6), with a mean absolute error of 111 mm (Table 2). These errors were propagated into the simulation with data assimilation without the forest strategy. Fortunately, the ratio between modeled SWE and synthetic SWE observations in forested grid cells and the nearest canopy-free grid cells had high levels of similarity. Therefore, estimating snow in forest regions using the nearest canopy-free pixels (Fig. 2) improved snow simulations significantly." I'm struggling a bit here to ensure that there is no impact based simply on the structure of the experiment. Forest SWE in the open loop simulation is too high. Assimilating the synthetic SWE retrievals lowers the forest SWE values so they agree better with the nature run, indicating a positive influence that the radar SWE product would deliver. But the synthetic radar data were generated from the nature run, perturbed with 20% random uncertainty. Because the nature run itself is used as the generator of the synthetic data and the reference, is this a case of the synthetic data adjusting the open loop back to itself? In this sense, the finding that the synthetic radar-retrieved SWE data is nudging the open loop back to the nature run does not prove the positive impact of the radar product but rather is just a mathematical adjustment achieved by using the nature run itself to ultimately adjust the open loop simulation. Presumably, prescribing a higher error value and considering radar-specific uncertainties on top of the 20% random error would result in less of an improvement. Conversely, applying a lower error value would result in more of an improvement. I think this issue of error characterization (see also comment #2 above), and independence of the results from the quality of the nature run itself needs to be addressed. Presumably the reduction in the forest SWE positive bias achieved via assimilation is what drives the reduction in positive bias in the streamflow estimate as well (line 346)?*

Thanks for the comment. You are correct that the data assimilation causes the open loop simulation to adjust the SWE towards the nature run in periods and locations without snowmelt. However, this study uses a "fraternal twin" setup wherein we use a different model for the nature run (SnowModel), and the open loop simulations (Noah-MP) with and without assimilation. This setup causes divergences in model states that are based on the difference in model physics (e.g., Kim et al., 2021; Mudryk et al., 2015), in addition to differences in forcing (Table S1). Therefore, we believe that this setup avoids the great majority of internal consistencies that the reviewer is expressing concerns about here. In fact, we chose a "fraternal twin" setup since "identical twin" OSSEs, which use the same

models, tend to overestimate the benefit of simulated observations (Yu et al., 2019). We now include a brief discussion on this in the main text (lines 182 - 187).

We believe that the results are representative of the degree to which we can expect to improve our large scale snow estimations provided radar SWE retrievals and the novel forest strategy. For example, the open loop and nature run simulations exhibit widespread and often uncorrelated differences in the volume, distribution and evolution of snow. Despite the fact that we don't change the model, forcing, or downscaling, our results show that assimilating SAR retrievals using a gap-based snow correction strategy in forested areas could improve common snow modeling errors provided only 10-14 day observation repeats and conservative assumptions about SAR retrieval capabilities in 1) forested grid cells and 2) grid cells experiencing snowmelt.

Finally, as we acknowledge earlier in our responses, SAR retrievals 1) could be subject to more systematic model errors based on terrain characteristics, and 2) will not have binary (full or none) observational capabilities over any 250m grid cell, but rather have retrieval accuracies that are a function of the distribution and density of multiple vegetation types, including forests and other surface vegetation (e.g., shrubs and tussocks). While it would be great to include these tests, this is still a topic of active research, making any physically based and location-specific approach for SAR retrieval degradations difficult to parameterize. Instead, this study uses conservative representations of SAR retrievals to demonstrate the value of extrapolating SAR-based snow corrections in some of the most difficult to observe landscapes. Using these simulations as a baseline, we are planning future research to focus on the landscape and sensor-specific issues that you mention here in smaller regions where these issues could be most prevalent (see the second-to-last paragraph of Section 4).

7. *Line 314: "mean SWE evolution tracked the nature run simulation significantly better than the open loop simulation in the spring snowmelt period." This is an interesting result. Essentially it shows that if you get peak SWE right, you can track SWE during the melt period even in the absence of many radar measurements (you had only 5% usable data during the melt period as stated on line 311). But I wonder how dependent this result is on the dynamics of the melt period. Presumably if there are additional snowfall events after peak SWE this may not be the case? Were the results in Figure 7 replicable for the other watersheds? And I wonder how replicable this result is from year to year (you would need to speculate here since this is outside the scope of this study)?*

Thanks so much. We were excited about this result. We discuss the modeled runoff in the last paragraph of Section 3, including the results shown in Figure 7 (Upper Colorado), the Great Basin (Figure S6), Pacific Northwest (Figure S7), and California (Figure S8). The NSE statistics for each of these regions and simulations are also reported in Table 1. To summarize, runoff improved with assimilation in both the Upper Colorado and Pacific Northwest for the simulation with data assimilation and the forest strategy. This wasn't surprising since a significant portion of the seasonal snow was in forested locations for these two domains, and the open loop simulation experienced widespread high biases in both SWE and runoff (relative to the nature run). However, the runoff accuracies for the

California and Great Basin domains changed very little. This was driven by the smaller changes to domain SWE volume with assimilation (relative to the Upper Colorado and Pacific Northwest). Because we used a fraternal twin OSSE with two different snow models, runoff performance could still be biased by the differences between models, even if SWE was largely improved. Here, the simulations using the Noah-MP snow model (OL, DA, and DA+F) tended to simulate less low-elevation winter snowmelt but more-rapid spring snowmelt, resulting in low-biased winter runoff and high-biased spring runoff relative to the nature run. Therefore, in these domains, the improvements to runoff from SWE assimilation were largely offset by the difference in snowmelt between the two snow models (Noah-MP versus SnowModel, Table S1).

We agree that abnormal dynamics in both the winter accumulation and spring snowmelt season could decrease the accuracy of simulations and make SAR observations more challenging. We added an acknowledgement of this in lines 487 - 493. In this study, 2019 had multiple spring snowfall events (e.g., Figure 7). However, despite this, the differences between spring SWE simulated by the nature run, open loop simulation, and simulations with data assimilation were driven much more by differences in winter snow accumulation and spring snowmelt rates.

Finally, while we agree that the inclusion of multiple years is outside of the scope of this particular study, we agree that a discussion of annual snow variability was missing from the manuscript. We now include this in lines 487 - 493. We also responded to similar comments about this from Reviewer #1 (question #3) and Reviewer #2 (question #2).

8. *The two major limitations of volume scattering based radar approaches are forest and wet snow. The focus is on forest in this study, but synthetic SWE under wet snow conditions was also masked. Can a similar strategy as was developed to fill in forest areas be used to address the wet snow challenge? This is out of scope to do this in this study, but perhaps something for the discussion?*

This is a great question, and one that would be interesting to investigate. We now acknowledge this in lines 493 - 495. However, there may be some additional challenges to making SWE estimates in regions experiencing snowmelt based on SAR observations from surrounding, unmelting regions. The forest approach presented here focuses primarily on winter periods, when the difference between nearby unmelting grid cells is due to differences in snow accumulation. At these spatial scales (~250m), model biases may be more likely to be driven by mesoscale-scale precipitation biases, which we hypothesize may have high spatial autocorrelation (lines 451 - 455). Contrary to this, differences in SWE between two grid cells, one experiencing snowmelt and the other not, could be driven by the same differences in SWE accumulation, in addition to differences from the cumulative snowmelt energy (influenced by processes like terrain shading, canopy shading and longwave emission, and temperature changes with elevation). We hypothesize that spatial differences in cumulative melt energy may exhibit less spatial autocorrelation, and particularly as the departure in snowmelt onset timing grows between grid cells.

*9. Line 19: 'popular' seems like an odd word choice to me...how about 'widely-used'?*

Good idea. We revised this in the updated abstract.

*10. Line 58: add the references for the limitations of passive microwave SWE datasets.*

Thanks for catching this. These references were updated (line 59).

*11. Line 60: I would specify that the Lievens et al study uses C-band SAR.*

Good idea. We specified this in our edits.

*12. Line 63: "To overcome these limitations, modeling and data assimilation systems are needed that can extend the coverage and utility of available measurements to areas, times, and variables that are not directly observed." Well said!*

Thank you!

*13. Lines 135-140: can some additional references be added for Noah-MP and TOPMODEL? I think Niu and Yang (2004) focuses only on the snow processes...*

We have a citation for Niu et al. (2011) earlier on when Noah-MP is first introduced in the manuscript (line 122). We also agree that we should have added a citation for TOPMODEL. It is now included (line 156)

*14. Figure 2: I find the dashed circle to be distracting and unnecessary. The gray shading shows the swath; the dashed line grid illustrates the SAR data. I don't know what the circle means...*

Thanks for the comment. We now see how this is confusing. The dashed circle was intended to represent the radius at which the unforested SWE corrections are sampled. For this example, the grid cell near the edge of the swath has fewer non-forested grid cells from which to derive the SWE adjustment since a portion of the search radius falls outside of the satellite swath. We revised the figure caption to make this clearer. We also revised the figure to show that we instead grabbed information from a 5-by-5 square window of grid cells.

References:

Durand, M., Johnson, J.T., Dechow, J., Tsang, L., Borah, F., Kim, E.J.: Retrieval of SWE from dual-frequency radar measurements: Using time series to overcome the need for accurate a priori information. EGUsphere 1 – 23. https://doi.org/10.5194/egusphere-2023-1653, 2023.

Errico, R.M., Yang, R., Masutani, M., Woollen, J.S.: The estimation of analysis error characteristics using an observation systems simulation experiment. Meteorologische Zeitschrift 16, 695–708, 2007.

Garnaud, C., Bélair, S., Carrera, M.L., Derksen, C., Bilodeau, B., Abrahamowicz, M., Gauthier, N., Vionnet, V.: Quantifying Snow Mass Mission Concept Trade-Offs Using an Observing System Simulation Experiment. Journal of Hydrometeorology 20, 155–173. https://doi.org/10.1175/JHM-D-17-0241.1, 2019.

Kim, R.S., Kumar, S., Vuyovich, C., Houser, P., Lundquist, J., Mudryk, L., Durand, M., Barros, A., Kim, E.J., Forman, B.A., Gutmann, E.D., Wrzesien, M.L., Garnaud, C., Sandells, M., Marshall, H.-P., Cristea, N., Pflug, J.M., Johnston, J., Cao, Y., Mocko, D., Wang, S.: Snow Ensemble Uncertainty Project (SEUP): quantification of snow water equivalent uncertainty across North America via ensemble land surface modeling. The Cryosphere 15, 771–791. https://doi.org/10.5194/tc-15-771-2021, 2021.

Mudryk, L.R., Derksen, C., Kushner, P.J., Brown, R.: Characterization of Northern Hemisphere Snow Water Equivalent Datasets, 1981–2010. Journal of Climate 28, 8037–8051. https://doi.org/10.1175/JCLI-D-15-0229.1, 2015.

Rott, H., Yueh, S. H., Cline, D. W., Duguay, C., Essery, R., Haas, C., Hélière, F., Kern, M., Macelloni, G., Malnes, E., Thompson, A.: Cold regions hydrology high-resolution observatory for snow and cold land processes. Proceedings of the IEEE, 98(5), 752-765, 2010.

Rott, H., Duguay, C., Etchevers, P., Essery, R., Hajnsek, I., Macelloni, G., Malnes, E., and Pulliainen, J.: CoReH2O Report for mission selection: An Earth Explorer to observe snow and ice, Tech. rep., European Space Agency, https://earth.esa.int/eogateway/documents/20142/37627/CoReH2O-Report-for-Mission-Selection-An-Earth-Explorer-to-observe-snow-and-ice.pdf, 2012.

Singh, S., Durand, M., Kim, E., Barros, A.P: Bayesian physical-statistical retrieval of snow water equivalent and snow depth from X- and Ku-band synthetic-aperture-radar demonstration using airborne SnowSAR in SnowEx17. EGUsphere 1 – 35. https://doi.org/10.5194/egusphere-2023-1987, 2023.

Tsang, L., Durand, M., Derksen, C., Barros, A.P., Kang, D.-H., Lievens, H., Marshall, H.-P., Zhu, J., Johnson, J., King, J., Lemmetyinen, J., Sandells, M., Rutter, N., Siqueira, P., Nolin, A., Osmanoglu, B., Vuyovich, C., Kim, E., Taylor, D., Merkouriadi, I., Brucker, L., Navari, M., Dumont, M., Kelly, R., Kim, R.S., Liao, T.-H., Borah, F., Xu, X.: Review article: Global monitoring of snow water equivalent using high-frequency radar remote sensing. The Cryosphere 16, 3531–3573. https://doi.org/10.5194/tc-16-3531-2022, 2022.

Yu, L., Fennel, K., Wang, B., Laurent, A., Thompson, K.R., Shay, L.K.: Evaluation of nonidentical versus identical twin approaches for observation impact assessments: an ensemble-Kalman-filter-based ocean assimilation application for the Gulf of Mexico. Ocean Science 15, 1801–1814. https://doi.org/10.5194/os-15-1801-2019, 2019.

Zhu, J., Tan, S., King, J., Derksen, C., Lemmetyinen, J., and Tsang, L.: Forward and inverse radar modeling of terrestrial snow using SnowSAR Data, IEEE Transactions on Geoscience and Remote Sensing, 56(12), 7122-7132, doi:10.1109/TGRS.2018.2848642, 2018.

Zhu, J., Tan, S., Tsang, L., Kang, D.K., and Kim, E.: Snow water equivalent retrieval using active and passive microwave observations. Water Resources Research 57, no. 7, 2021.

---

## Author Response (AR2)

We would like to thank the anonymous reviewer for agreeing to review this manuscript again. We were pleased to hear that they found the results and text to be clearer. Below, we include in-line responses to each of the reviewer's comments.

1. *The revised introduction is now much clear in that the characteristics of the SWE retrievals used within the OSSE are hypothetical. But I think one additional point should still be made – missions in development (both past missions like CoReH20 and current missions like TSMM) have SWE retrieval requirements ranging from approximately 20-30%. So in that sense the uncertainty you apply in this study is realistic and appropriate. But it is still to be seen if these missions could actually deliver retrievals that meet this requirement … so using line 94 (from the tracked changes version) as an example, what do you think about revising the text to read: "… what is the added utility of spaceborne active remote sensing SWE information (assuming retrievals meet currently defined mission requirements) across the western U.S. and Canada?" This would just make it explicitly clear that reproducing these OSSE results using 'real' retrievals requires the algorithm development teams to achieve the stated mission requirements.*

   Thanks for the suggestion. We agree that we should reiterate that since these retrievals are hypothetical, the results are subject to an instrument and mission that meet the requirements laid out in the text. We liked your suggestion about adding a parenthetical to the framing questions at the end of the introduction. This can be found in line 87.

2. *Line 113: change 'snowy' to 'snow covered'*

   This was changed in the updated text (line 103).

3. *Line 211: I would not say that Ku-band measurements have a hard sensitivity limit of 200 mm of SWE, rather that uncertainty is expected to be higher in deep snow conditions.*

   Great point! We revised this sentence to make it clearer that measurements are not capped at 200 mm SWE, but instead increase in uncertainty as snow gets deeper (lines 190 – 192). We felt that "deep snow" was subjective, so we still included "≥ 200 mm" in a parenthetical for reference.